# Intelligent Food Packaging: Quaternary Ammonium Chitosan/Gelatin Blended Films Enriched with Blueberry Anthocyanin-Derived Cyanidin for Shrimp and Milk Freshness Monitoring

**DOI:** 10.3390/foods13142237

**Published:** 2024-07-16

**Authors:** Dan Chen, Jialiang Lv, Ao Wang, Huimin Yong, Jun Liu

**Affiliations:** College of Food Science and Engineering, Yangzhou University, Yangzhou 225127, China; dchen@yzu.edu.cn (D.C.); mx120211254@stu.yzu.edu.cn (J.L.); mz120222097@stu.yzu.edu.cn (A.W.); 008694@yzu.edu.cn (H.Y.)

**Keywords:** color-indicator, pH-sensitive, responsive packaging, blueberry anthocyanin, milk monitoring

## Abstract

Blueberry anthocyanin-derived cyanidin (BAC) was used to prepare a series of responsive food freshness packaging films by compounding it with quaternary chitosan (QC) and gelatin (G). The fundamental properties, pH sensitivity, and functional attributes of the films were examined. The BAC solutions exhibited notable variations in color (from red to pink to violet) under different pH conditions. The incorporation of BAC resulted in improved UV–vis shielding capabilities but compromised the mechanical strength of the films (with tensile strength values from 85.02 to 44.89 MPa, elongation at break from 13.08% to 3.6%, and water vapor transmission rates from 5.24 × 10^−9^ to 7.80 × 10^−9^ g m^−1^ s^−1^ Pa^−1^). The QC-G-BAC films, containing 5–15 wt% BAC, exhibited noticeable color changes in acidic/ammonia environments within a short timeframe, easily discernible to the naked eye. Furthermore, the inclusion of BAC significantly enhanced the antioxidant and antimicrobial properties of the films. The addition of 5–15 wt% BAC to QC-G-BAC films could be employed for assessing the freshness of fresh shrimp (from red to dark red) and pasteurized milk (from red to dark earthy yellow). Among them, the total color difference (ΔE) of QC-G-BAC5 film was significantly correlated with the pH, acidity, and total colony count of pasteurized milk (R = 0.846, −0.930, −0.908, respectively). This new concept in smart packaging offers a straightforward and user-friendly freshness indicator.

## 1. Introduction

Packaging is essential in the food industry for preserving the quality and safety of food, as well as prolonging its shelf life [1]. Generally, safety and quality changes occur while storing, distributing, and transporting food. The date printed on the package is commonly utilized by consumers to assess the freshness and quality of packaged foods. Nevertheless, for certain products, like milk and specific meats, the expiration date alone may not be adequate for determining their freshness and quality [2]. Smart color indicator packages represent an innovative and intelligent approach to detecting, tracking, and safeguarding food quality. These indicators are utilized to signal the condition of food products, allowing consumers to assess their quality and freshness in real time within a smart packaged environment [3]. The color changes exhibited by the indicator during the storage of packaged food serve as a visual assurance of quality and its impact on food safety [4]. This is achieved through the principle that microbial metabolites react with the indicator during microbial growth on packaged foods, resulting in discernible color changes due to alterations in pH value. Consequently, these markers allow consumers to comprehend the freshness or quality of a food item based on chemical transformations or microbial growth throughout its distribution process [5].

Anthocyanin, a natural extract that exhibits distinct color changes at varying pH levels, has been employed in creating smart pH indicator films in recent times [2,6]. Over 600 anthocyanins have been discovered in the natural world, and they are predominantly found in higher plants while being notably absent from lower plants with the exception of certain mosses and ferns [7]. The diverse array of hues present in a variety of flowers, fruits, and vegetables, including carmine, indigo, magenta, purple, and pink, can be explained by the existence of anthocyanins [6]. These substances exhibit pH-dependent color-changing properties and also serve as bioactive agents with anti-angiogenic, free radical scavenging, anticancer, antidiabetic, antimicrobial, and neuroprotective effects [2,7,8]. Anthocyanins have been utilized in functional foods, traditional medicines, and food additives. As a class of polyphenolic flavonoids that function as natural colorants reflecting red to blue light within the visible spectrum, the color-changing characteristics of anthocyanins hold significant implications for monitoring food quality and shelf-life determination while also enhancing consumer interest in food products, ultimately serving as valuable indicators for food packaging design [9].

In the past few years, there has been an increasing recognition of the food freshness monitoring capabilities of blueberry anthocyanins when incorporated into films [10]. Research have been concentrated on enhancing the stability of blueberry anthocyanins for use in intelligent color-developing packaging materials, such as those used for tilapia, through microencapsulation with carboxymethyl starch/xanthan gum composite and acylation with maleic acid [11,12,13]. The primary anthocyanidins found in blueberries include peonidin, malvidin, pelargonidin, cyanidin, delphinidin, and petunidin [14]. The objective of this research was to incorporate blueberry anthocyanins-derived cyanidin (BAC) into a hybrid film matrix consisting of quaternary chitosan (QC)/Gelatin (G). The structural, physical, and functional characteristics of the resulting film was evaluated. It was then used to monitor the freshness of shrimp and pasteurized milk (the main foods for protein supplementation for Chinese consumers), contributing to the creation of an innovative multi-purpose food packaging film.

## 2. Materials and Methods

### 2.1. Materials and Reagents

BAC (CAS: 13306-05-3, formula: C_15_H_11_O_6_, formula weight: 287.244) was procured from Macklin Biochemicals Ltd. (Shanghai, China) (the structure is shown in Figure 1A). QC (N-(2-hydroxy) propyl-3-trimethylammonium chitosan chloride with 83% substitution and an average molecular weight of 250 kDa), as well as 2,2-diphenyl-1-picrylhydrazyl (DPPH), were obtained from Macklin Biochemicals Ltd. in Shanghai, China. Gelatin sourced from cold water fish skin with an average molecular weight of 60 kDa and an isoelectric point of 6 was obtained from Sigma Chemical Company in St. Louis, MO, USA. The Food Microbial Laboratory of Yangzhou University (Yangzhou, China) generously provided the foodborne pathogens used in this study (*Escherichia coli* ATCC 43895, *Salmonella typhimurium* ATCC 14028, *Staphylococcus aureus* ATCC 6538, and *Listeria monocytogenes* ATCC 19115). Freshwater shrimp (*Macrobrachium rosenbergii*) and pasteurized milk were acquired from a local Yonghui supermarket in Yangzhou, China. All other reagents utilized were of analytical grade.

### 2.2. pH Sensitivity of BAC Solution

The pH susceptibility of BAC solution was assessed by dissolving 1 mg of BAC powder in 10 mL of various buffers (ranging from pH 3 to 12) for a duration of 30 min [15]. The appearances and visible absorption spectra of the resulting BAC solutions were, respectively, measured using a photometer and a Lambda 35 spectrophotometer (PerkinElmer Ltd., Hopkinton, MA, USA) within the wavelength range of 450 to 700 nm.

### 2.3. Preparation of BAC-Based pH-Responsive Films

A series of QC-G-BAC films were prepared using the method outlined by Hu et al. [16]. Briefly, varying quantities of BAC (0 g, 0.17 g, 0.34 g, and 0.51 g), 1.7 g of QC, and 1.7 g of G were dissolved in 160 mL of distilled water at a temperature of 50 °C for a duration of 2 h to form a film-forming matrix containing 0%, 5%, 10%, and 15% wt/wt of BAC. Subsequently, every film-forming matrix was mixed with 0.68 g of glycerol, degassed, and poured into a 24 cm × 24 cm Plexiglas plate. These mixtures were dried at a temperature of 30 °C under a relative humidity of 50% for 3 days to obtain the films named as QC-G-BAC0 film, QC-G-BAC5 film, QC-G-BAC10 film, and QC-G-BAC15 film, respectively. These films were then stored in a desiccator at 20 °C under a relative humidity of 50%.

### 2.4. Structural Characterization of the Films 

#### 2.4.1. Scanning Electron Microscopy (SEM) of the Films

The film samples were subjected to rapid freezing in liquid nitrogen to break them, and then affixed to aluminum rods. Subsequently, the film samples were coated with a layer of gold and examined using a GeminiSEM 300 (Carl Zeiss, Oberkochen, Germany) at an accelerating voltage of 5 kV and a magnification of 800× for observation of the cross-sections.

#### 2.4.2. Fourier Transform Infrared (FT-IR) and X-ray Diffraction (XRD) Spectra of the Films

The FT-IR spectra of the films were obtained using a Varian 670 spectrophotometer (Varian Corp., Palo Alto, CA, USA) with an attenuated total reflection in the range of 400–4000 cm^−1^. The spectra were collected with 16 scans and a resolution of 2 cm^−1^. The XRD test was performed using the D8 Advance diffractometer (Bruker AXS GmbH, Karlsruhe, Germany), with a 2θ data collected in the range of 4° to 50° and a speed of 4°/min.

### 2.5. Physical and Chemical Characteristics of the Films

#### 2.5.1. Optical Characteristics

The films’ color parameters (L, luminance/brightness; a, red/green; b, yellow/blue) were assessed using a SC-80C colorimeter from Beijing Kangguang Instruments Co., Ltd. (Beijing, China). The measurements were taken against a white reference plate background. Subsequently, the total color difference (ΔE) of the films was then determined using the following approach [15]:(1)$$\Delta E=\sqrt{{\left(L*\;-\;L\right)}^{2}+\;{\left(a*\;-\;a\right)}^{2}+\;{\left(b*-\;b\right)}^{2}}$$

The white reference plate’s color parameters were described as L* (93.66), a* (−0.94), and b* (−0.14).

#### 2.5.2. Thickness and Mechanical Characteristics

The thickness of each film was measured using a Mitutoyo No. 293-766 digital micrometer (Sanyo Corporation tester, Tokyo, Japan) with an accuracy of 0.001 mm. Each film was cut to dimensions of 6 cm × 1 cm and firmly attached to the jig of a TMS-Pro texture analyzer from Virginia Food Technology (Blacksburg, VA, USA) at a crosshead speed of 6 cm/min and an initial clamping distance of 4 cm [16]. The tensile strength (TS) and elongation at break (EAB) were then calculated to determine the mechanical characteristics of the films. Equations (2) and (3) are as follows:(2)$$\mathrm{TS}\;(\mathrm{MPa})=\frac{F}{\left(\chi \times W\right)}$$
(3)$$\mathrm{EAB}\;\left(\%\right)=\frac{\Delta L}{L_{0}}\times 100$$
where F denotes the stress at film fracture (in newtons), χ represents the thickness of the film (in millimeters), W indicates width of the film (in millimeters), ΔL signifies the increased length at film rupture (in millimeters), and L_0_ represents the initial length of the film (in millimeters).

#### 2.5.3. Water Vapor Permeability (WVP)

Each film was enclosed in a 15 mL centrifuge tube with an inner diameter of 1.5 cm, containing 9 g of dry silica gel. The tubes were then stored in a desiccator with distilled water to maintain 100% relative humidity at 20 °C for a period of 7 days. The weight of the tube was recorded daily [16].
(4)$${\mathrm{WVP}\;(g\cdot m}^{-1}{\cdot s}^{-1}{\cdot \mathrm{Pa}}^{-1})=\frac{W\;\times \;\chi }{t\;\times \;A\;\times \;\Delta P}$$
where W represented the added weight of the centrifuge tube (in grams), χ denoted the thickness of the film (in meters), t signified the duration (in seconds), A indicated the permeable area of the film (in square meters), and ΔP referred to the saturated vapor pressure at 20 °C.

#### 2.5.4. Water Solubility

Each film was cut into 2.5 cm × 2.5 mm pieces and dried in a drying oven at 105  °C for 24 h to obtain the initial weight of the dried films. Subsequently, the dried samples were soaked in distilled water at 25  °C and visually inspected for dissolution to determine the time required to obtain complete solubility. Any undissolved films after 1 h were removed and re-dried in the oven at 105  °C for another 24 h to obtain their remaining dry weight. The water solubility of the films was then calculated using the following specific formula [17]:(5)$$\mathrm{Water}\;\mathrm{Solubility}\;(\%)=\frac{W_{i}-W_{r}}{W_{i}}\times 100$$
where $W_{i}$ denotes the initial dry weight and $W_{r}$ denotes the remaining dry weight of the films.

#### 2.5.5. Thermogravimetric Analysis (TGA)

TGA was conducted using a TGA8000 instrument (PerkinElmer, Waltham, MA, USA) with a temperature range from 30 to 800 °C. Film samples (2 mg) were examined in a nitrogen environment at a flow rate of 20 mL/min. The differential thermogravimetric curve (DTG) was obtained by taking first-order derivatives of the thermogravimetric curve.

#### 2.5.6. Volatile Acid and Ammonia Sensitivity

The films’ responses to volatile acid and ammonia were evaluated using a previously established method [18] with some adjustments. The 2.5 cm × 2.5 cm film samples were separately attached to the headspace of a sealed container containing 15 mL of 98% acetic acid solution or 1 mol/L ammonia water at a temperature of 20 °C. The change in the film’s color was regularly monitored using an optical scanner (Perfection V370, Seiko Epson Co., Nagano, Japan).

### 2.6. Functional Activities of the Films

#### 2.6.1. Antioxidant

The films’ ability to neutralize free radicals was evaluated using the DPPH radical scavenging method. Each film sample was completely submerged in a methanolic solution containing 100 μmol/L of DPPH for one hour at 20 °C in the absence of light, and then the absorbance of the reaction solution at 517 nm was measured [16].

#### 2.6.2. Antimicrobial 

The film’s effectiveness against *E. coli* ATCC 43895, *S. typhimurium* ATCC 14028, *S. aureus* ATCC 6538, and *L. monocytogenes* ATCC 19115 were assessed using the agar diffusion method [16]. In brief, a bacterial culture (0.2 mL) with a concentration of 10^6^ CFU/mL was uniformly distributed over a lysogeny broth agar plate. Subsequently, a circular film sample with a diameter of 9 mm was positioned on the plate’s surface and kept at 37 °C for 16 h. Finally, the size of the bacteriostatic zone surrounding the film sample was measured using a vernier caliper.

### 2.7. Application in Visualized Food Freshness Monitoring

#### 2.7.1. Monitoring Shrimp Freshness

The freshness of *P. rosenbergii* shrimps was assessed by monitoring the films [16]. Film samples were affixed to the inner lid of a sealed plastic container with 30 g of fresh shrimp and then placed in storage at a temperature of 20 °C for 48 h. The total volatile basic nitrogen (TVB-N) content in the shrimp was determined using a K9860 Kjeldahl distillation apparatus from Jinan Hanon Instruments Co., Ltd., located in Jinan, China. At the same time, any color changes in the film samples were recorded using a Perfection V370 optical scanner from Nagano Seiko Epson in Japan.

#### 2.7.2. Monitoring Pasteurized Milk Freshness

The 20 mL milk sample was transferred with caution into a plastic Petri dish measuring 90 mm in diameter and with a rim depth of 15 mm. The dish was then placed in a temperature-controlled incubator set at 25 °C for the purpose of cultivation [19]. During different time intervals (0, 24, 48, and 72 h), a film measuring 1 cm × 2 cm was immersed in milk for 30 s. The relationship (Pearson correlation coefficient (R)) between film color change (ΔE) and milk spoilage trend was investigated with respect to pH, titratable acidity (°T), and total colony counts (Log_10_ CFU/mL) at the corresponding time intervals. Total colony counts were determined using the plate count agar technique. The plates were incubated under aerobic conditions at 37 ± 1 °C for a period of 24 h to count bacterial colonies [20]. 

### 2.8. Biodegradability of the Films

A film sample measuring 2.0 cm × 2.5 cm was weighed and placed in gauze and then into a clay pot with moist and fertile soil with a temperature in the range of 20 ± 2 °C, which was collected from the campus of Yangzhou University. The film was observed after 12 days and the biodegradability was assessed by measuring the reduction in weight according to the following specific formula [21]: (6)$$\mathrm{Biodegradability}\;(\%)=\frac{\omega_{o}-\omega_{i}}{\omega_{0}}\times 100$$
where $\omega_{0}$ denotes the initial weight and $\omega_{i}$ denotes the remaining weight of the films.

### 2.9. Statistical Analysis

The mean ± SD was used to present the data. Statistical analysis was performed using one-way analysis of variance (ANOVA) and Duncan’s multiple range test with a significance level set at *p* < 0.05, utilizing SPSS 22.0 software (SPSS Inc., Chicago, IL, USA).

## 3. Results and Discussion

### 3.1. pH Sensitivity of BAC Solution

The color development of BAC solution under varying pH conditions was illustrated in Figure 1B,C. It was evident from Figure 1B that the hue of the solution at pH 3–6 leans towards red, with a tendency to lighten as the pH increases. Conversely, the solution at pH 8–12 exhibits a purple hue, transitioning from light to dark as the pH rises. At pH 7, anthocyanin displays a color between red and purple. The color difference in BAC solution across various pH solutions could be attributed to molecular changes leading to the formation of new chromogenic substances [22]. In Figure 1B, it is clear that the solution shows a peak absorption at a wavelength between 520–532 nm within the pH range of 3–6. As the pH increased, the absorbance decreased, consistent with the observed trend of lighter color in Figure 1C. At pH levels of 8–11, there was a shift in the maximum absorption wavelength to 579 nm, corresponding to a gradual darkening of the solution color, as depicted in Figure 1B. At pH 7, the maximum visible absorption wavelength was measured at 550 nm, aligning with the purple–red hue observed in Figure 1B. Notably, at a pH level of 12, no absorption peak for the BAC solution was detected due to the complete alteration in the chemical structure of cyanidin. Comparable findings have been documented for alternative anthocyanins, including those derived from amaranth extract [16]. The color-changing properties of BAC solution have important implications for monitoring food quality and determining shelf-life, serving as valuable indicators for the design of food packaging.

### 3.2. Structural Characterization of the QC-G-BAC Films 

#### 3.2.1. SEM Analysis

Figure 2 displays SEM images of the cross-section of a series of QC-G-BAC films. The cross-section of QC-G-BAC 0 film appeared smooth and uniform, showing no pores or cracks, indicating good mixing and the compatibility of all membrane components (QC, G, and glycerol), consistent with the SEM images observed by Hu et al. [16]. When BAC was added, the QC-G-BAC5 and QC-G-BAC10 films were smooth and dense in cross-section, indicating good compatibility of BAC with QC-G matrix. Upon increasing the BAC content to 15 wt%, a bark-like grain structure emerged in the cross-section of the QC-G-BAC15 film. This phenomenon was attributed to a decrease in compatibility among the membrane components caused by cyanidin [6]. Hu et al. [16] and Ramziia et al. [23] also reported that the cross-sections of chitosan-G films and QC-G films became inhomogeneous after the addition of proanthocyanidins and anthocyanidin extracts, respectively. The findings indicated that the BAC concentration had a notable impact on the structural characteristics of the films.

#### 3.2.2. FT-IR and XRD Analysis

FT-IR spectroscopy has the capability to investigate the functional groups and intermolecular interactions present in thin films. As depicted in Figure 3A, QC-G-BAC0 film showed stretching at 3279 cm^−1^ (corresponding to N-H and O-H), 2928 cm^−1^ (indicative of C-H), 1638 cm^−1^ (representing C=O), and 1542 cm^−1^ (associated with N-H and C-N) [16,23]. The band at 1240 cm^−1^ was identified as the methylene group of glycine in G [24] and the characteristic band at 1477 cm^−1^ was linked to the C-H bending of the trimethylammonium group of QC [25]. Similar observations were made in chitosan-G-based films [23]. BAC addition resulted in distinctive band stretching of phenolics at 1028 cm^−1^ (C-O-C) [23,26]. It is worth mentioning that the addition of BAC changed the band strength of the films. The broadening and enhancement of the band near 3284 cm^−1^ and weakening of the stretching near 1544 cm^−1^ were due to hydrogen bonding interaction between BAC and the film matrix. When betaine-rich amaranth extract was added to the QC-G film, the same significant band enhancement was observed [16].

The crystal structures of the different films are shown in Figure 3B. The QC-G-BAC0 film showed a broader diffraction peak of QC at 20.33°, which is in agreement with the findings of Tharun et al. [27] since the chitosan quaternization disrupts the intramolecular hydrogen bonding. After the addition of BAC to the matrix, some changes in the diffraction peaks occurred, as evidenced by the appearance of diffraction peaks at 8.21°, 11.19°, 15.97°, 17.99°, and 22.81°. The increase in the crystallinity of the QC-G-BAC films and the reduction in their amorphous region are attributed to the hydrogen bonds formed between BAC and the QC-G matrix. Similarly, Shi et al. [28] demonstrated that cyanidin-3-glucoside enhanced the crystallinity of the bacterial cellulose matrix through hydrogen bonding (2θ = 14° and 23°). 

### 3.3. Physical and Chemical Characteristics of the QC-G-BAC Films

#### 3.3.1. Color and Light Transmittance

As shown in Figure 4A, the transparency and colorlessness of the QC-G-BAC0 film resulted from the colorless nature of its membrane components, namely QC, G, and glycerol [16]. After the addition of BAC, the QC-G-BAC films showed varying degrees of pink to dark red coloring. The color parameters for the QC-G-BAC films are presented in Table 1. Compared with the QC-G-BAC0 film, the a, b, and ΔE values of the QC-G-BAC5, QC-G-BAC10, and QC-G-BAC15 films gradually increased, and the L value gradually decreased, which was consistent with the gradual deepening of the reddish–yellow color of the films. These findings suggest that BAC content has an impact on the color of QC-G-BAC films. Similar changes in color were observed in anthocyanin-rich acacia bean gum/poly(vinyl alcohol) films and ovalbumin/propylene glycol alginate nanocomplex films [9].

Food packaging films that possess the capability to block light are able to prevent some UV–visible rays from reaching the food products, thus helping to inhibit oxidation and deterioration [29]. As depicted in Figure 4B, the QC-G-BAC0 film exhibits a high level of light transmittance, similar to that of the chitosan-G film [23]. The light transmittance of the QC-G-BAC films exhibited a gradual decrease as the BAC content increased, indicating that acoustic emission significantly enhances the UV–visible blocking capability of the films. This enhancement was due to the absorption of UV–visible radiation by unsaturated bonds (e.g., C=C, C=N and C=O) present in BAC [30,31]. Furthermore, the presence of BAC results in a decrease in the transmission of light through the film due to light scattering or reflection. Previous studies have also reported that red cabbage sourced cyanidin improved the UV–visible barrier properties of polyacrylonitrile conjugated films [32].

#### 3.3.2. Thickness, WVP, Water Solubility, and Mechanical Properties

As depicted in Table 2, the thickness of the QC-G-BAC films exhibited an increase with higher BAC content, ranging from 0.045 mm (QC-G-BAC0 film) to 0.065 mm (QC-G-BAC15 film), attributed to the incorporation of BAC into the film network, leading to an elevation in solids content within the film. Furthermore, the introduction of anthocyanin-rich extracts into chitosan-G or QC-G films also resulted in an augmentation of film thickness [16]. 

The capacity of these films to impede moisture transfer is commonly assessed by their WVP. Generally, it is desirable for food packaging films to possess a low WVP in order to reduce the transfer of moisture between the food and its environment, thereby prolonging the food product’s shelf life [33]. The WVP of a film is affected by different elements, such as the thickness of the film, the quality of the film structure, and the connections between functional groups within the components of the film [33]. The addition of BAC resulted in a notable rise in the WVP value of the QC-G-BAC films, suggesting that BAC has the potential to improve the water vapor transmission characteristics of these films. This increase in WVP value can be attributed to the formation of a lamellar texture path by BAC within the film matrix, which speeds up the movement of water vapor.

The potential applications of a film are affected by its water solubility. In certain situations, it is preferable to use films that are not soluble in water in order to enhance the water resistance, integrity, and shelf life of food products [3]. On the other hand, there are cases where it is beneficial to use soluble films that dissolve in water before consumption for various purposes, such as food encapsulation or coating [19]. The water solubility of the QC-G-BAC films created was tested by immersing them in distilled water at 25 °C for 1 h, and the results can be seen in Table 3, showing close to 100% solubility in all instances. Similar results were found with more acetylated sweet potato starch-based films, with a water solubility of 99.16% [17]. These findings indicate that QC-G-BAC films could be widely used as a packaging material for short-term food contact indication and cleaning before consumption.

The TS and EAB of the film are crucial indicators for its performance in protecting food during transportation and storage [17]. The addition of BAC significantly decreased both TS and EAB, suggesting that anthocyanins may disrupt the interaction between gelatin and chitosan, leading to a reduction in cross-linking. This is consistent with SEM results and is also similar as a previous finding on betalains’ impact on chitosan-G mixtures [16]. However, a study reported no significant difference between TS (20.45–25.31 MPa) and EAB (40.78–48.75%) both when chitosan/gum arabic films contained or did not contain anthocyanin (3 mg/mL)[34]. Furthermore, TS and EAB initially increased (1–5 wt%) and then decreased (5–7 wt%) in wheat gluten protein/apple pectin films supplemented with blueberry anthocyanin extract (1–7 wt%) [35]. In another study, reduced TS and EAB of films containing an excess of sweet potato extract (20 wt%) was observed [36].

#### 3.3.3. Thermal Characteristics

The TGA–DTG curves are depicted in Figure 5. The initial phase (30–150 °C) is associated with the release of water, while the following phase (151–245 °C) is linked to the evaporation of glycerol [37]. During the third stage (246–800 °C), BAC, QC, and G undergo depolymerization and decomposition. It is worth noting that the films experience the greatest weight loss at around 310 °C. The relatively slower weight loss observed in the QC-G-BAC films suggests a slight enhancement in their thermal stability, possibly due to interactions between BAC and the QC-G matrix, thereby mitigating thermal degradation. Similarly, it was found that blueberry anthocyanin extract has a significant effect on the thermal stability of wheat gluten protein and apple pectin-based film [35], and *Ipomoea coccinea* extract significantly affects the thermal stability of polyvinyl alcohol and the guar gum matrix [36].

#### 3.3.4. Volatile Acid Discoloration and Ammonia Sensitivity Analysis

In order to evaluate the pH responsiveness of the films, the QC-G-BAC films were subjected to acetic acid and ammonia gasses, and the results are presented in Figure 6. The films’ red color became stronger when exposed to volatile acetic acid gas, due to the transformation of BAC from a pink quinone anhydrous group to a red–yellow ion. Upon exposure to ammonia gas, the membrane rapidly transitioned to a green color within 10 min and then turned olive after 20 min. The occurrence can be ascribed to the penetration of ammonia into the film matrix followed by hydrolysis leading to the generation of hydroxyl ions, thereby creating an alkaline environment that triggers the structural transformation of BAC from a carbinol pseudo-base state to a chalcone state [38]. The above findings suggest that the QC-G-BAC (5–15 wt%) films exhibited high sensitivity to fluctuations in pH levels, with a 5 wt% BAC content being sufficient to accurately reflect changes in environmental pH. Furthermore, it was observed that various matrix (chitosan/agarose, etc.) films containing anthocyanins displayed varying degrees of discoloration and response times in acidic and alkaline environments. These differences can be attributed to variations in the composition of anthocyanins as well as differences in testing conditions, including gas concentration and duration [2]. The findings demonstrate the possibility of using QC-G-BAC films as pH indicators to monitor the freshness of food.

### 3.4. Functional Properties of the QC-G-BAC Films

#### 3.4.1. Antioxidant 

The DPPH radical scavenging assay was used to evaluate the antioxidant activity of QC-G-BAC films (Figure 7A). It was observed that QC-G-BAC0 film showed the least DPPH radical scavenging activity when tested at concentrations between 1 and 5 mg/mL. The DPPH radical scavenging activity of QC-G-BAC films was significantly improved with an increase in BAC content. Significantly, there were no observable variances in the DPPH radical scavenging capability between QC-G-BAC10 film and QC-G-BAC15 film when tested at levels of 4 and 5 mg/mL. The heightened DPPH radical scavenging activity observed in QC-G-BAC films is mainly due to the ability of BAC to scavenge free radicals. Previous studies have also highlighted the robust antioxidant properties of blueberry anthocyanin, with evidence suggesting that incorporating blueberry anthocyanin-enriched extract enhances film antioxidant capacity [10,11]. Consequently, it is conceivable that QC-G-BAC films may serve as effective antioxidant packaging for food products, thereby contributing to a prolonged shelf life.

#### 3.4.2. Antimicrobial 

The effectiveness of QC-G-BAC films in inhibiting microbial growth was evaluated using the agar diffusion technique. As depicted in Figure 7B, it can be observed that QC-G-BAC films with a BAC content ranging from 5–15 wt% displayed notable antimicrobial effects against four foodborne pathogens. The antimicrobial effect of QC-G-BAC0 film was primarily attributed to the presence of positively charged trimethylammonium groups in QC. These groups are able to form electrostatic interactions with microbial cells that carry a negative charge, leading to the denaturation of proteins and disruption of cellular components [14,15,39]. The antimicrobial activity of the QC-G-BAC5-15 films was found to be increased in comparison to the QC-G-BAC0 film, suggesting that the inclusion of BAC improved the effectiveness of the films against microorganisms. Lozano-Navarro et al. [40] demonstrated that the growth of *Penicillium notatum*, *Aspergillus niger*, *Aspergillus fumigatus*, and aerobic mesophilic bacteria was effectively suppressed by blueberry anthocyanin extracts. Notably, *E. coli* ATCC 43895 and *L. monocytogenes* ATCC 19115 exhibited enhanced susceptibility to the QC-G-BAC5 film. The film’s antimicrobial effectiveness against the four foodborne pathogens showed no significant variance at 10 wt% and 15 wt% BAC content. These findings indicated that the QC-G-BAC films possess antimicrobial activity at concentrations exceeding 5 wt%, albeit in a non-dose-dependent manner.

### 3.5. Biodegradability of the Films

The QC-G-BAC films buried in soil for 12 days all exhibited physical changes and weight loss (Figure 8). Upon statistical analysis, the difference in biodegradability of the films was statistically significant (*p* < 0.05). As shown in Figure 8, the biodegradation rates of the films ranged from 63.62 ± 0.41% to 51.12 ± 0.76%, which is comparable to the biodegradation rates (46.45–64.42%) of Mosambi (*Citrus limetta*) peel and Sago based films stored for 15 days as reported by Ahmad et al. [21]. Notably, no difference in biodegradability was observed between QC-G-BAC10 film and QC-G-BAC15 film, the reason for this item deserves further investigation.

### 3.6. Application of the Films

#### 3.6.1. Monitoring the Quality of Shrimp

The presence of TVB-N, which consists of ammonia, dimethylamine, and trimethylamine, is responsible for the deterioration in shrimp quality [16]. The sensitivity of volatile ammonia was utilized to monitor the freshness of shrimp (*M. rosenbergii*) by employing QC-G-BAC films. The levels of TVB-N in shrimp showed a gradual increase during the storage period, as shown in Table 3. After being stored for 3 days, the TVB-N level in shrimp was found to be 21.70 ± 0.99 mg/100 g, slightly above the limit set by the Chinese standard GB 2733-2015 limit (20 mg/100 g) [41]. It is noteworthy that both QC-G-BAC5 and QC-G-BAC10 films exhibited color changes to black and red between days 2 and 3, indicating their suitability for assessing shrimp freshness. In contrast, the color change for QC-G-BAC15 film did not occur until day 5 due to its higher BAC content. A recent study also demonstrated that films incorporating blueberry anthocyanins into wheat gluten and apple pectin could be utilized for monitoring shrimp freshness [35].

#### 3.6.2. Monitoring the Freshness of Pasteurized Milk

There is an immediate requirement to identify spoilage in perishable food items with a short shelf life (e.g., pasteurized milk) in real-time. Milk is susceptible to deterioration and acidification while being stored, which has led us to examine the durability and suitability of QC-G-BAC films for monitoring milk freshness (Table 4). In this study, the shelf life of milk was monitored using pH, acidity, and total colony count as quality parameters. The fresh milk started with a pH of 6.75 and an acidity level of 15.5 °T. As storage time increased, the pH dropped quickly, and the acidity rose significantly. The acidity of milk reached 18.5 °T after 24 h, and exceeded 28 °T after 48 h, surpassing quality regulations for pasteurized milk in China (GB 19645-2010) [42]. Additionally, the milk samples showed a colony count of over 10^7^ CFU/mL after 48 h, suggesting spoilage [43]. The color indicator was capable of distinguishing between fresh milk in the initial and final stages of spoilage as pH and acidity levels rapidly increase, causing a noticeable shift in label color from red to a natural yellow. This finding highlighted the indicator’s potential to accurately detect milk spoilage progression. Previous research has incorporated blueberry anthocyanins, blood orange anthocyanins, or shikonin to create smart packaging for prolonging the shelf life of milk [20,34,44]. Their findings indicated that the film’s color transitioned from a deep purple or dark blue to a lighter shade of purple or blue, which could potentially impede consumers’ ability to distinguish between them visually. In contrast to these studies, the shift from red to yellow in this study was more pronounced and visually discernible. 

Furthermore, analyzing the correlation between the TVB-N, overall number of colonies, pH, acidity, and ΔE of the films provided valuable information on the potential application of the prepared films for assessing shrimp and milk freshness. As indicated in Table 5, there was a strong positive relationship between the ΔE of QC-G-BAC films (5–15 wt%) and TVB-N level during the shrimp storage. Furthermore, there was also a strong positive relationship between ΔE and pH value, while a significant negative association was observed with total colony count during the pasteurized milk storage. Among them, the 5 wt% BAC-added QC-G-BAC film demonstrated simultaneous and significant correlations with pH, total colony counts, and acidity. Hence, certain factors, like pH, overall acidity, and total colony count, have shown promising potential for utilizing QC-G-BAC5 in the assessment of milk freshness. Similarly, Moazami Goodarzi et al. developed a simple colorimetric pH indicator using starch and carrot anthocyanins to evaluate the shelf life of milk [19].

## 4. Conclusions

In this investigation, innovative QC/G multifunctional films were successfully fabricated through the incorporation of BAC. The presence of BAC greatly improved the films’ ability to block UV–visible light, their sensitivity to volatile acids and ammonia-induced color changes, and their functional (antioxidant and antimicrobial) properties. The QC-G-BAC films (5–15 wt%) demonstrated potential for monitoring the freshness of fresh shrimp (from red to dark red) and milk deterioration (from red to yellow-green), leveraging their high visual recognition ability. From the results of the Pearson correlation, films with 5 wt% BAC addition were the best for milk quality indication, making it a promising candidate for further utilization in the food industry. 

## Figures and Tables

**Figure 1 foods-13-02237-f001:**
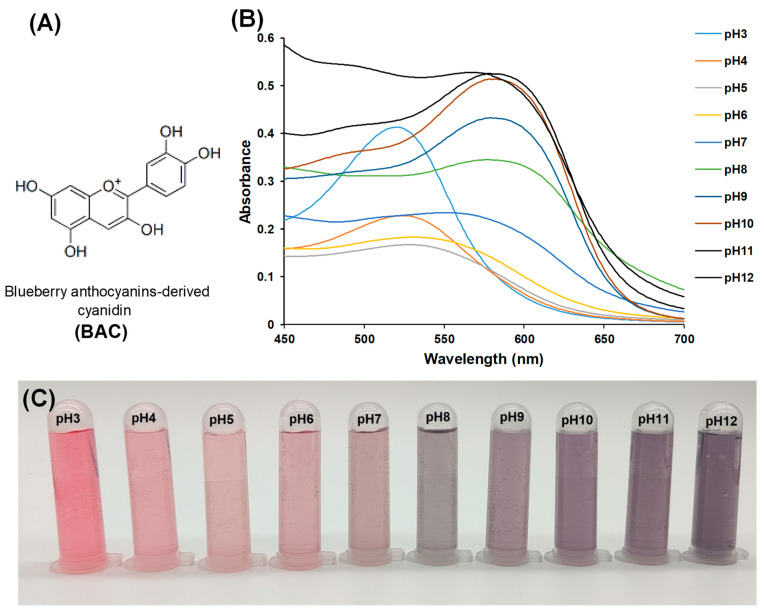
Structure of blueberry anthocyanin-derived cyanidin (BAC) (**A**) and visible absorption spectra (**B**) and color (**C**) of BAC at different pH.

**Figure 2 foods-13-02237-f002:**
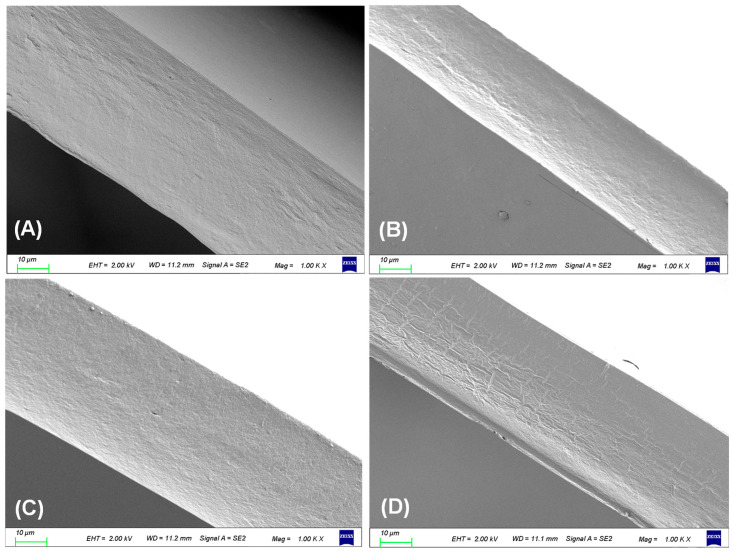
Scanning electron microscope (SEM) graphs of QC-G-BAC0 (**A**), QC-G-BAC5 (**B**), QC-G-BAC10 (**C**), and QC-G-BAC15 (**D**) film sections.

**Figure 3 foods-13-02237-f003:**
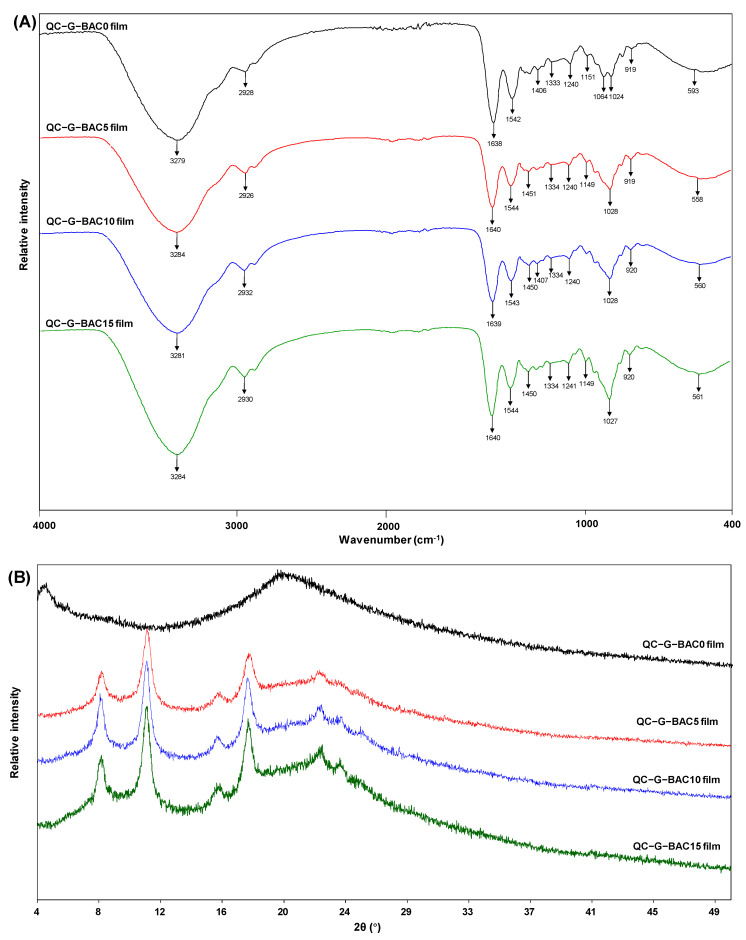
Fourier transform infrared (FT-IR) (**A**) and X-ray diffraction (XRD) (**B**) spectrogram of QC-G-BAC films. BAC, blueberry anthocyanin-derived cyanidin; G, gelatin; QC, quaternary chitosan.

**Figure 4 foods-13-02237-f004:**
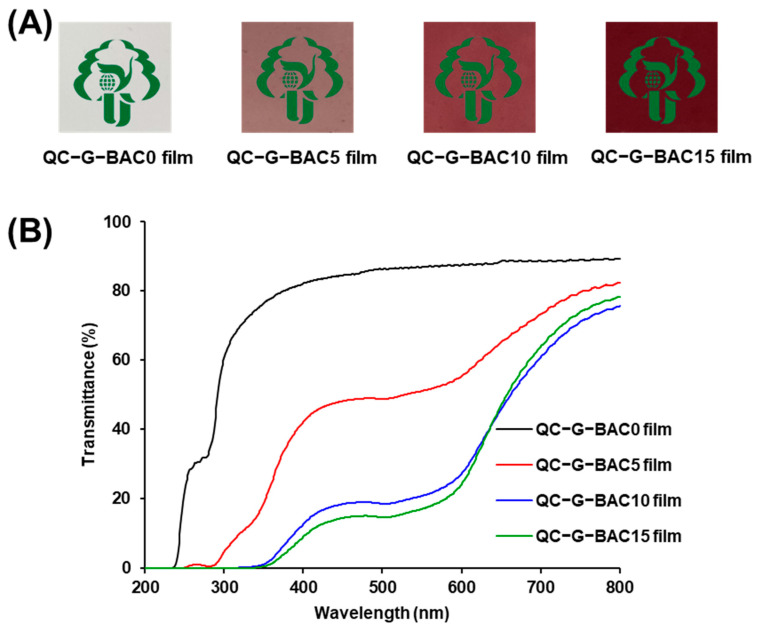
Physical appearance (**A**) and UV–visible transmittance (**B**) of QC-G-BAC films. BAC, blueberry anthocyanin-derived cyanidin; G, gelatin; QC, quaternary chitosan.

**Figure 5 foods-13-02237-f005:**
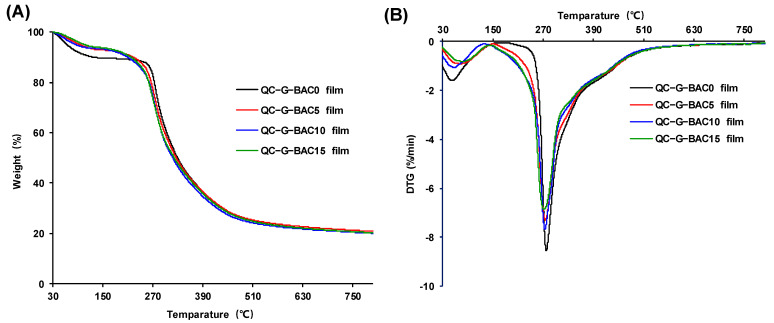
Thermogravimetric analysis (TGA) (**A**) and differential thermogravimetric (DTG) (**B**) curves of QC-G-BAC films. BAC, blueberry anthocyanin-derived cyanidin; G, gelatin; QC, quaternary chitosan.

**Figure 6 foods-13-02237-f006:**
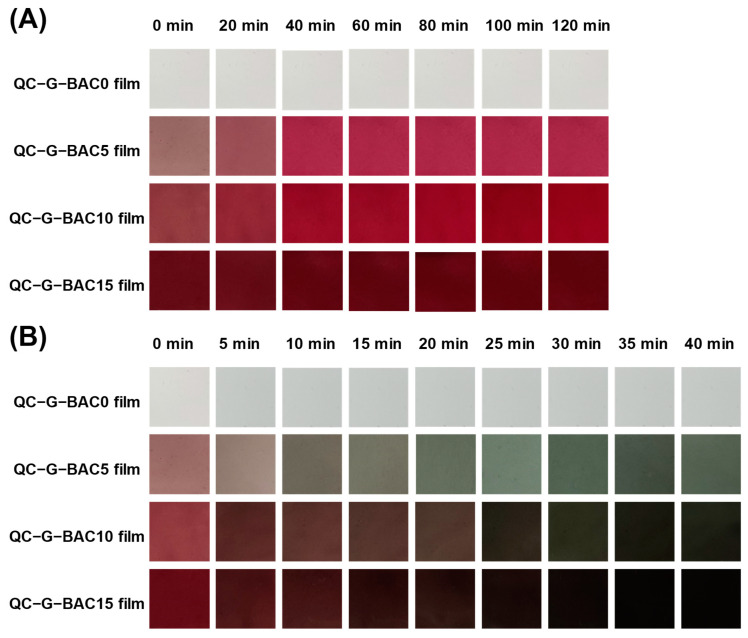
Volatile acid (**A**) and volatile ammonia (**B**) susceptibility of QC-G-BAC films. BAC, blueberry anthocyanin-derived cyanidin; G, gelatin; QC, quaternary chitosan.

**Figure 7 foods-13-02237-f007:**
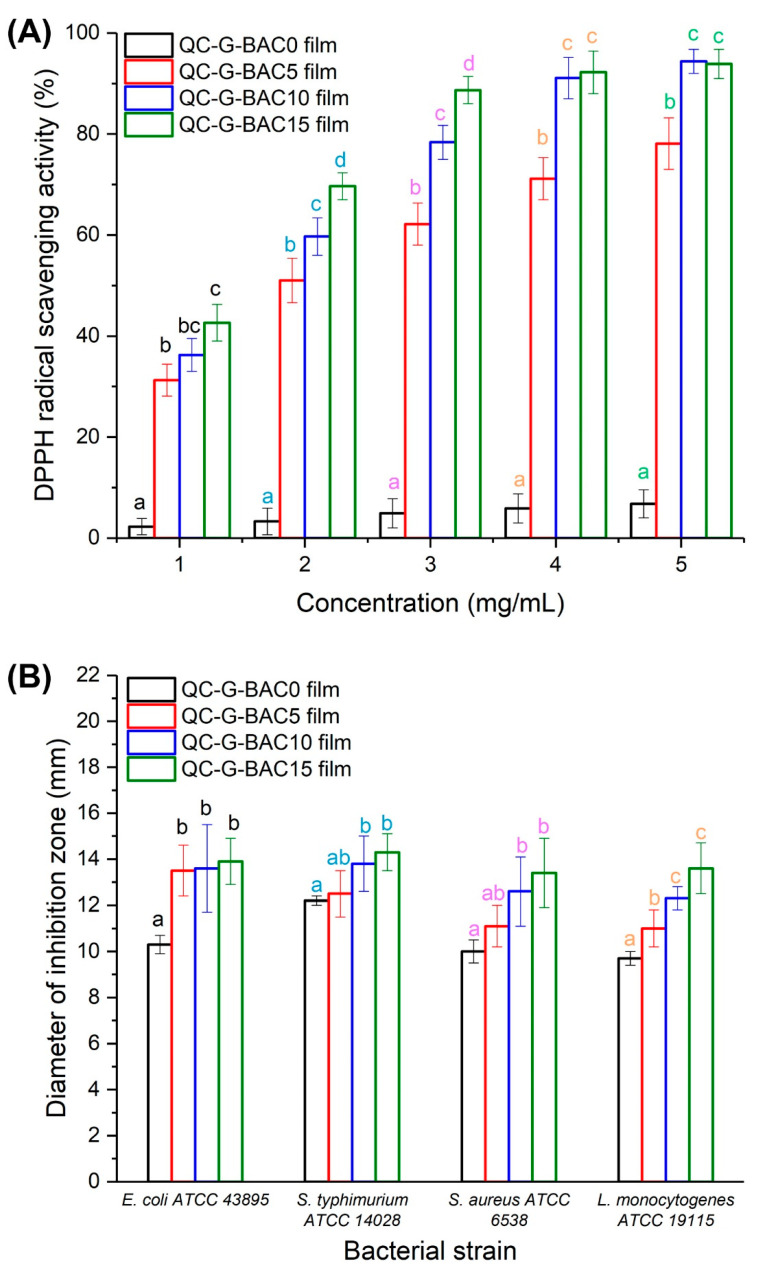
DPPH scavenging activity (**A**) and antimicrobial activity (**B**) of QC-G-BAC films. Each value represents the mean ± SD of triplicates. BAC, blueberry anthocyanin-derived cyanidin; G, gelatin; QC, quaternary chitosan. Different letters in the same color indicate significantly different values between the data (*p* < 0.05).

**Figure 8 foods-13-02237-f008:**
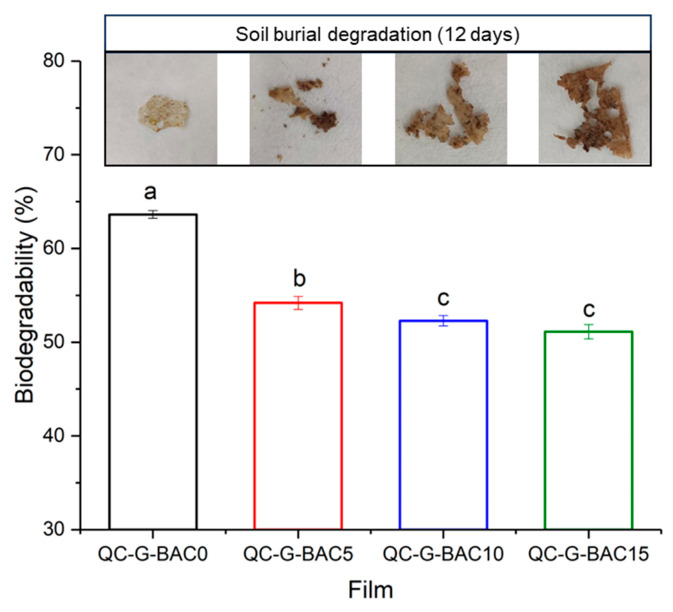
Biodegradability of QC-G-BAC films. Each value represents the mean ± SD of triplicates. BAC, blueberry anthocyanin-derived cyanidin; G, gelatin; QC, quaternary chitosan. Different letters in the same color indicate significantly different values between the data (*p* < 0.05).

**Table 1 foods-13-02237-t001:** Color values of QC-G-BAC films.

Film	L	a	b	ΔE
QC-G-BAC0 film	89.25 ± 0.61 ^a^*	0.57 ± 0.06 ^d^	−2.50 ± 0.03 ^d^	4.83 ± 0.04 ^d^
QC-G-BAC5 film	61.90 ± 1.00 ^b^	9.46 ± 0.02 ^c^	1.29 ± 0.30 ^c^	33.39 ± 0.99 ^c^
QC-G-BAC10 film	42.27 ± 0.22 ^c^	19.70 ± 0.38 ^b^	4.56 ± 0.27 ^b^	55.50 ± 0.12 ^b^
QC-G-BAC15 film	35.90 ± 0.01 ^d^	24.06 ± 0.04 ^a^	7.99 ± 0.04 ^a^	63.52 ± 0.10 ^a^

BAC, blueberry anthocyanin-derived cyanidin; G, gelatin; QC, quaternary chitosan. Values are given as mean ± SD (*n* = 3). * Different letters in the same column indicate significantly different values (*p* < 0.05).

**Table 2 foods-13-02237-t002:** Physical characteristics including thickness, WVP, TS, EAB, and water solubility of QC-G-BAC films.

Film	Thickness (mm)	WVP (×10^−9^ g m^−1^ s^−1^ Pa^−1^)	Water Solubility (%)	TS (MPa)	EAB (%)
QC-G-BAC0 film	0.045 ± 0.003 ^d^*	5.24 ± 0.08 ^b^	99.85 ± 0.47 ^a^	85.02 ± 3.38 ^a^	13.08 ± 0.03 ^a^
QC-G-BAC5 film	0.059 ± 0.004 ^c^	7.60 ± 0.97 ^a^	99.95 ± 0.66 ^a^	63.07 ± 1.83 ^b^	12.55 ± 1.61 ^a^
QC-G-BAC10 film	0.062 ± 0.002 ^b^	7.73 ± 0.72 ^a^	99.91 ± 0.58 ^a^	56.26 ± 1.96 ^c^	3.60 ± 0.25 ^b^
QC-G-BAC15 film	0.065 ± 0.003 ^a^	7.80 ± 0.98 ^a^	99.25 ± 0.98 ^a^	44.89 ± 3.83 ^d^	3.07 ± 0.52 ^b^

BAC, blueberry anthocyanin-derived cyanidin; G, gelatin; QC, quaternary chitosan. Values are given as mean ± SD (*n* = 3). * Different letters in the same column indicate significantly different (*p* < 0.05).

**Table 3 foods-13-02237-t003:** Changes in the TVB-N level of shrimp and the color of QC-G-BAC films during shrimp storage.

Time (Day)	TVB-N Level (mg/100 g)	QC-G-BAC0 Film		QC-G-BAC5 Film		QC-G-BAC10 Film		QC-G-BAC15 Film	
		Appearance	ΔE	Appearance	ΔE	Appearance	ΔE	Appearance	ΔE
0	2.10 ± 0.70 ^f^*	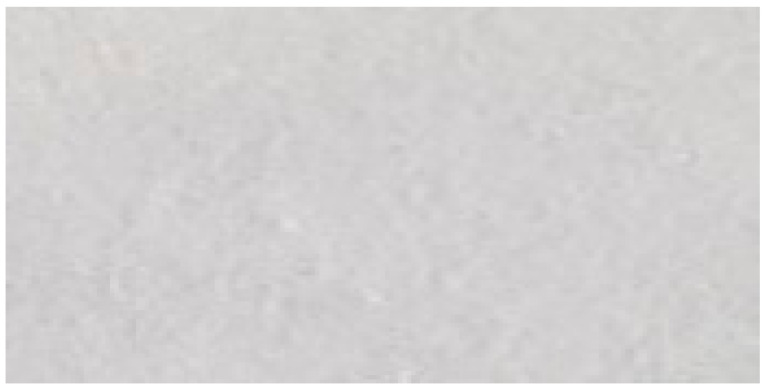	4.83 ± 0.04 ^a^	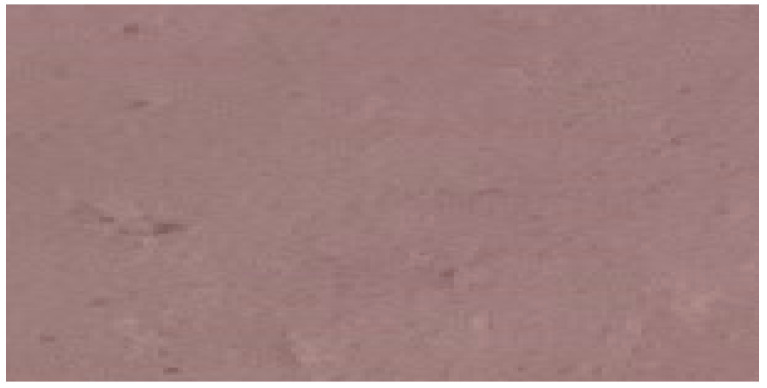	33.39 ± 0.99 ^f^	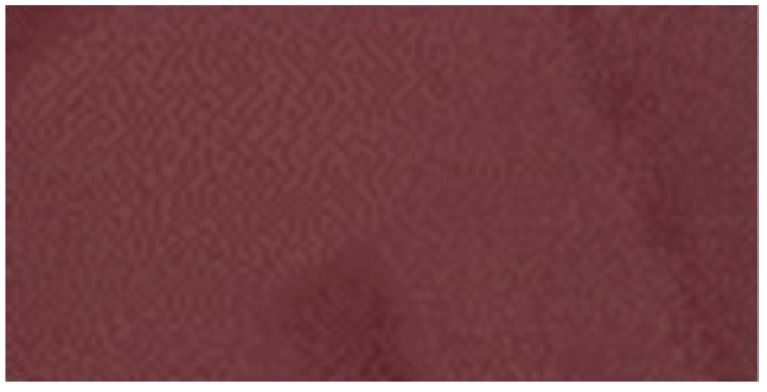	55.50 ± 0.12 ^g^	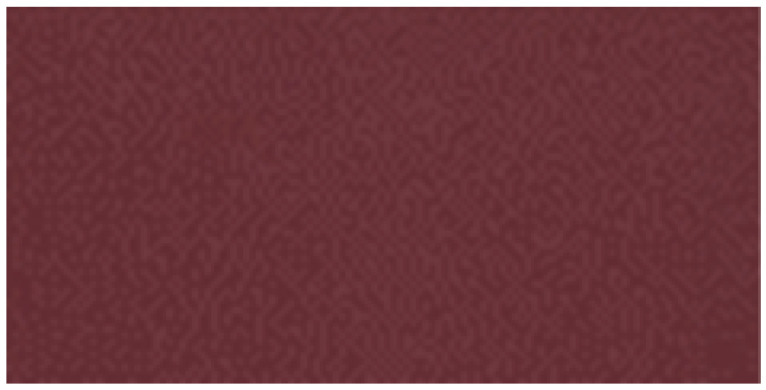	63.52 ± 0.10 ^e^
1	5.95 ± 0.49 ^f^	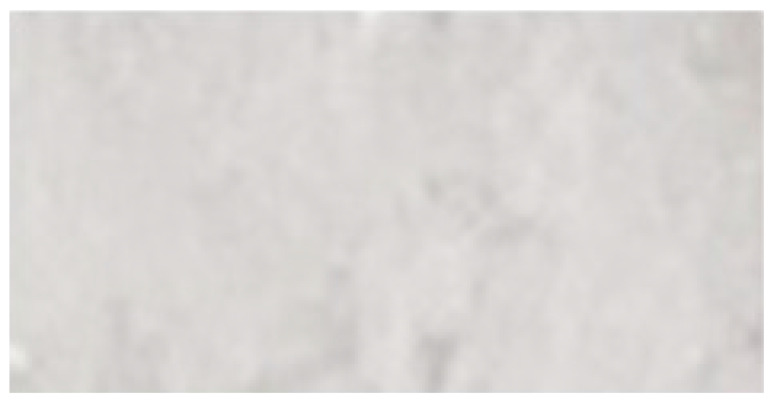	5.39 ± 0.63 ^a^	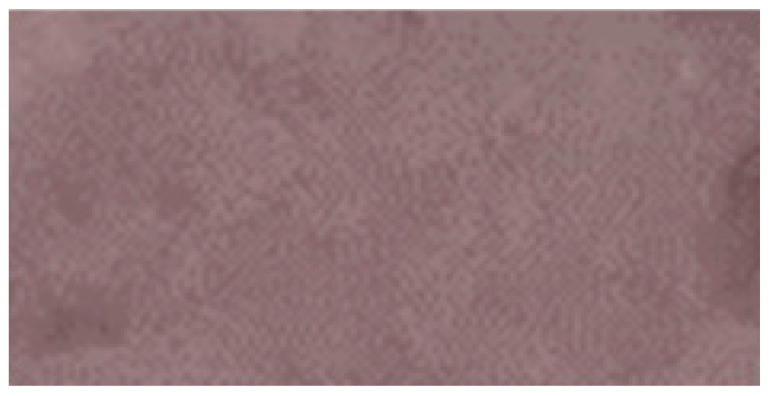	38.17 ± 0.75 ^e^	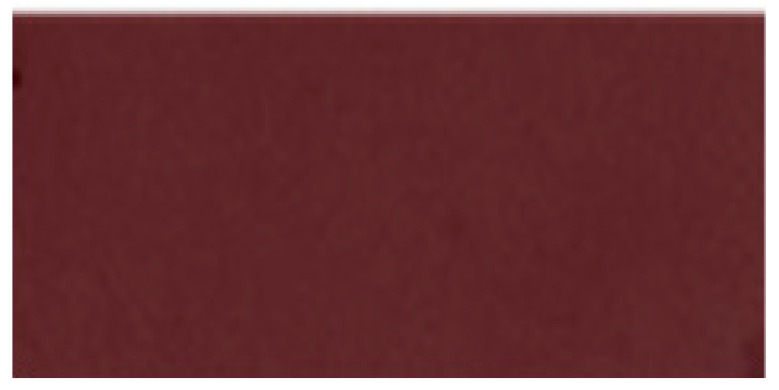	58.49 ± 0.41 ^f^	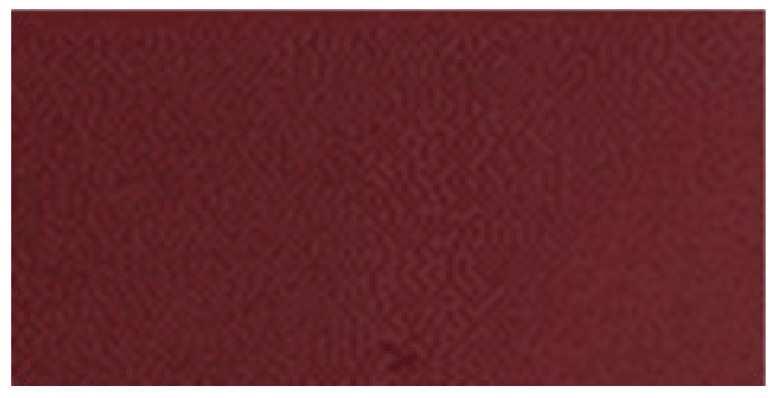	69.01 ± 1.17 ^d^
2	16.10 ± 0.99 ^e^	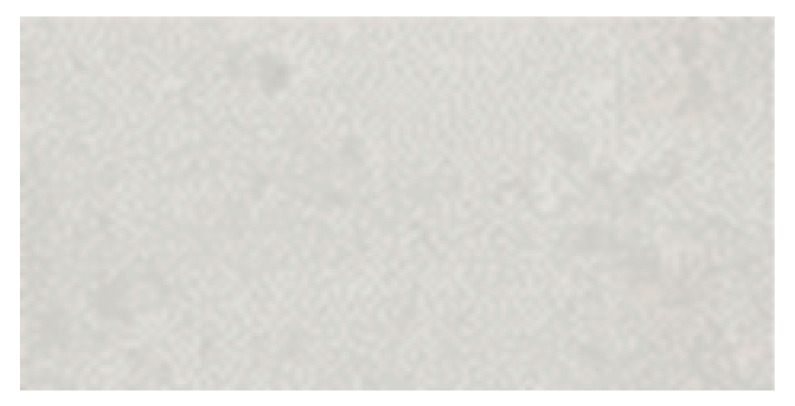	5.80 ± 0.60 ^a^	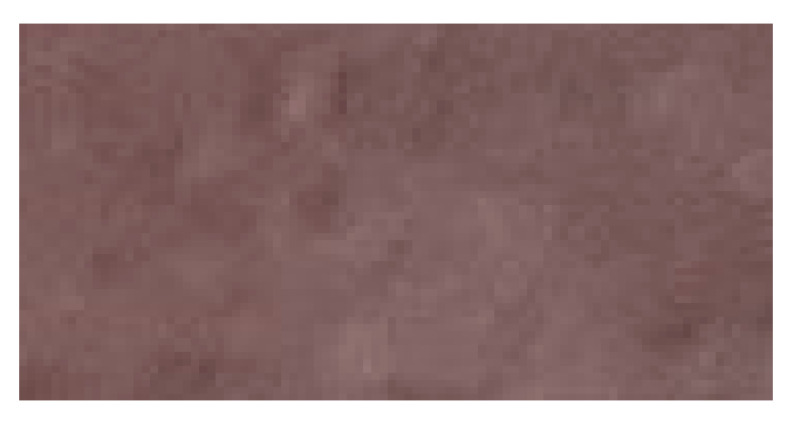	42.54 ± 0.90 ^d^	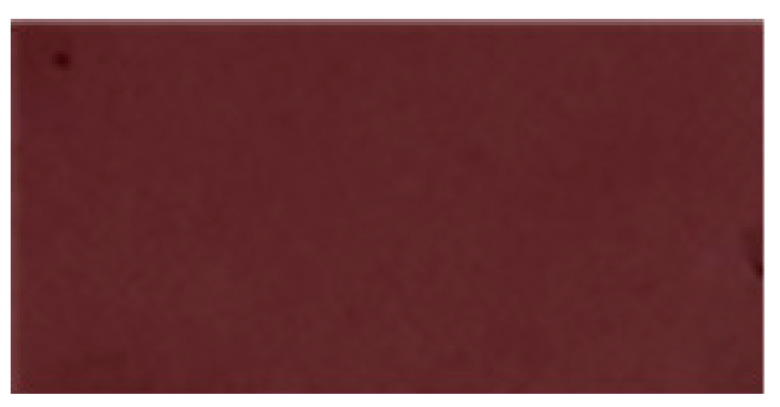	64.29 ± 1.38 ^e^	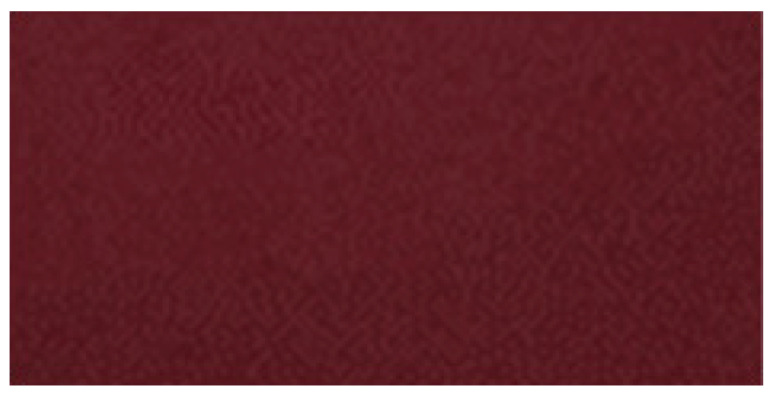	71.81 ± 0.58 ^c^
3	21.70 ± 0.99 ^d^	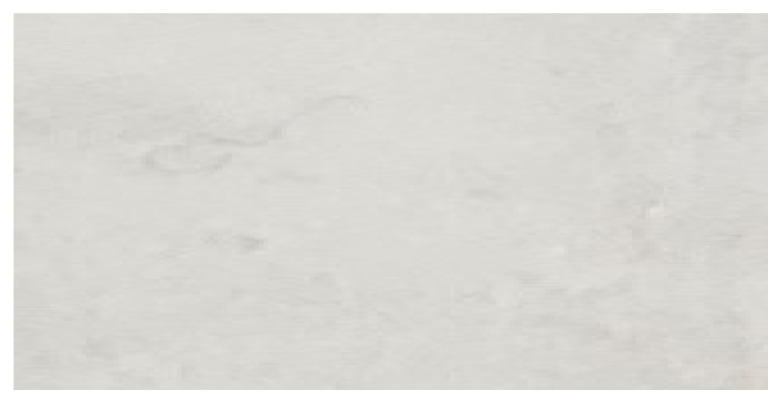	5.45 ± 0.68 ^a^	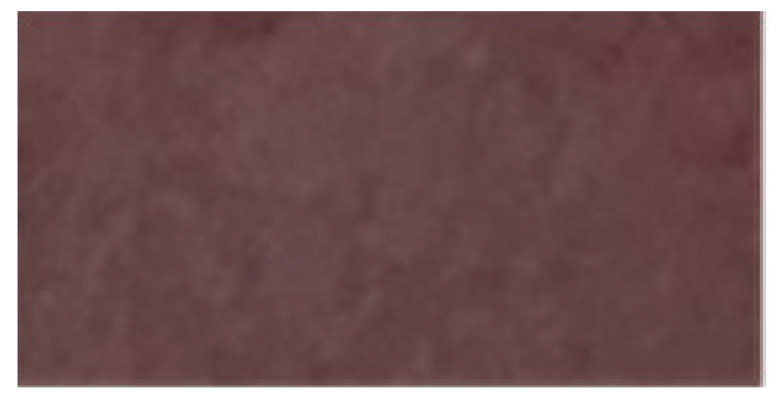	49.67 ± 1.82 ^c^	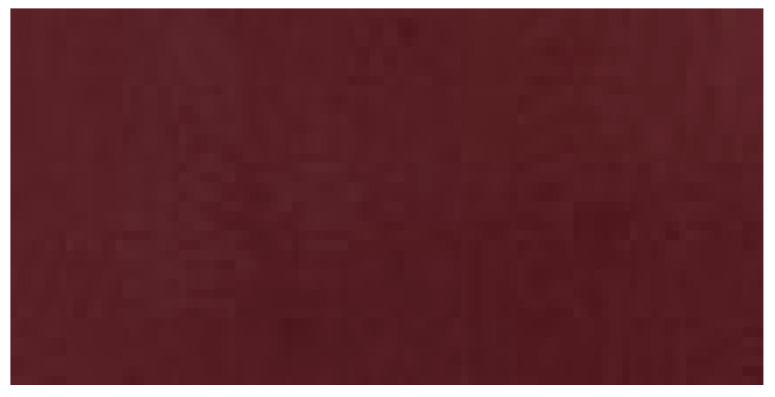	67.78 ± 0.80 ^d^	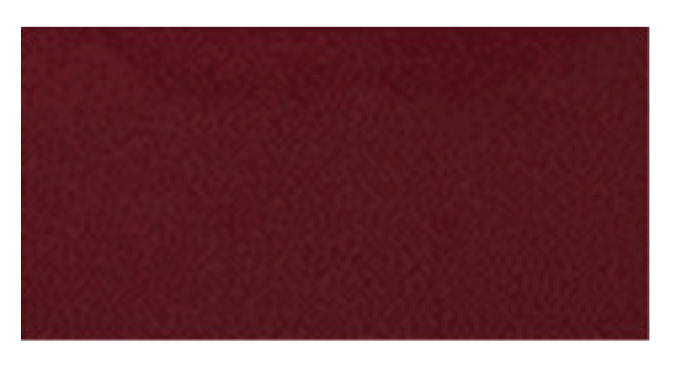	73.30 ± 1.45 ^bc^
4	28.70 ± 0.99 ^c^	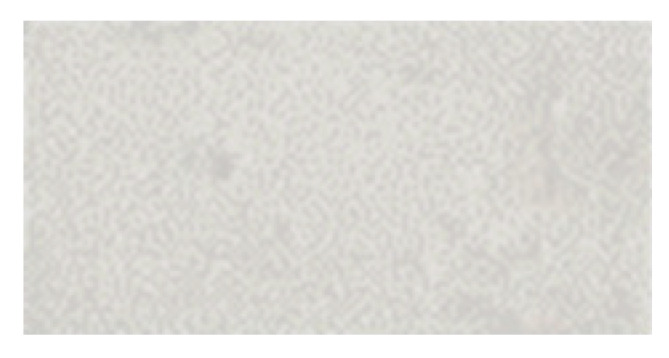	5.28 ± 0.62 ^a^	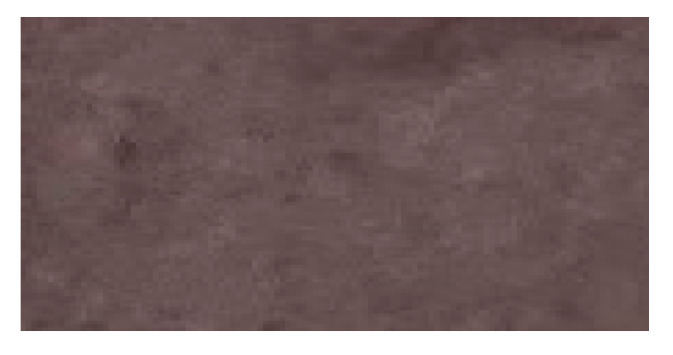	52.54 ± 1.51 ^c^	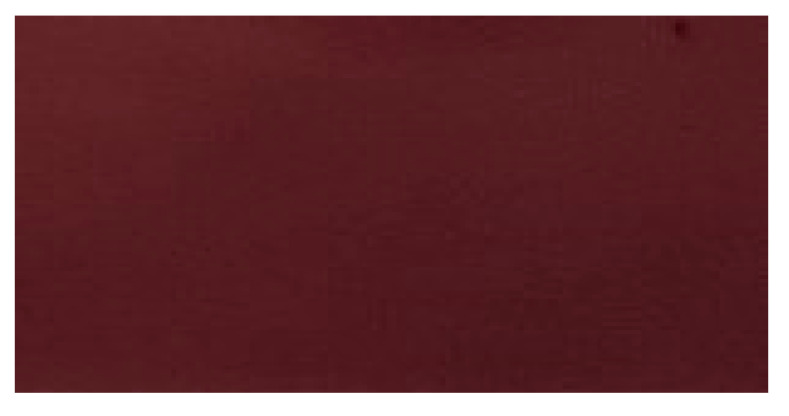	73.33 ± 2.23 ^c^	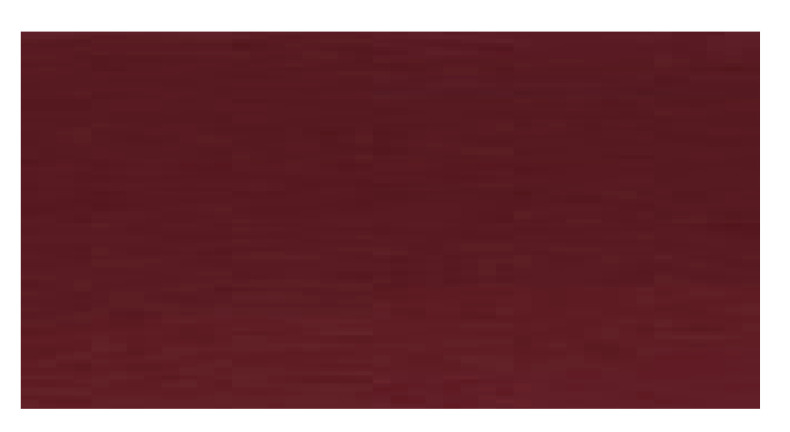	74.82 ± 0.76 ^b^
5	38.50 ± 2.97 ^b^	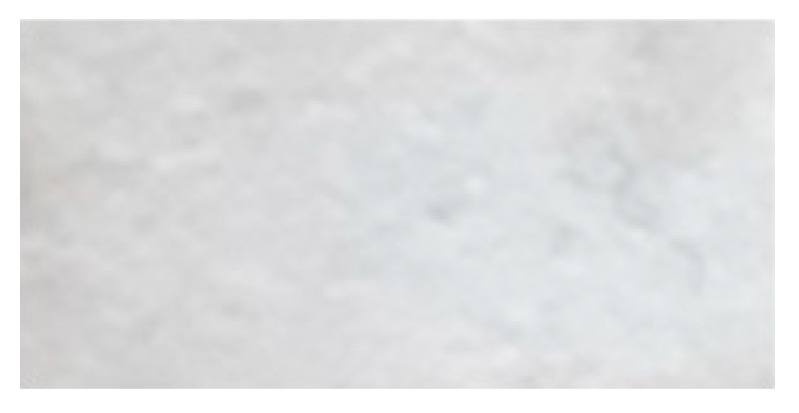	5.01± 0.72 ^a^	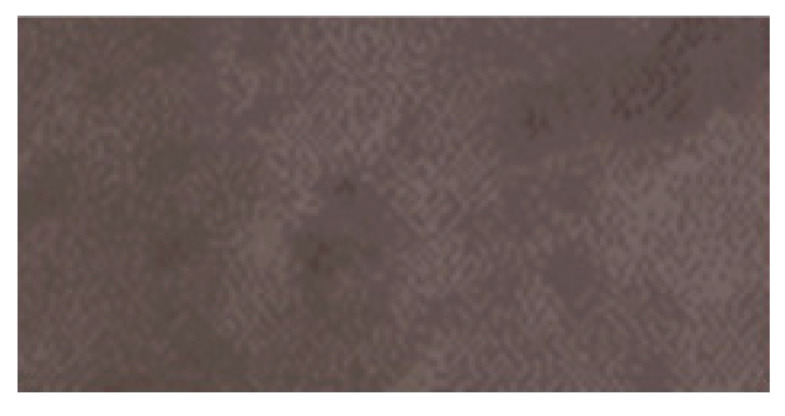	57.19 ± 1.48 ^b^	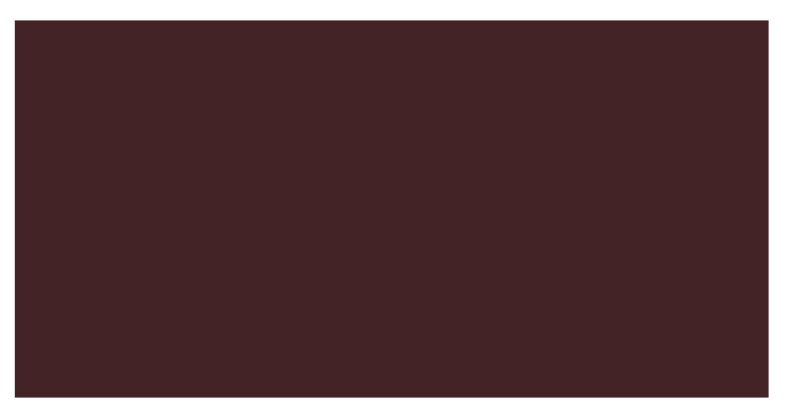	76.27 ± 0.32 ^b^	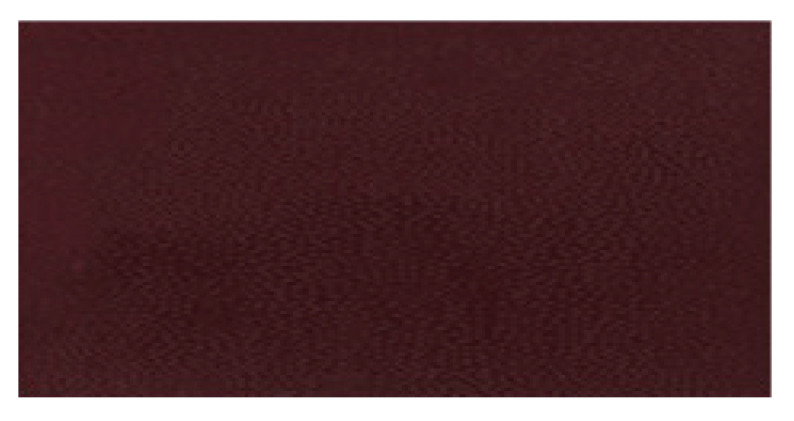	80.05 ± 1.23 ^a^
6	46.55 ± 3.46 ^a^	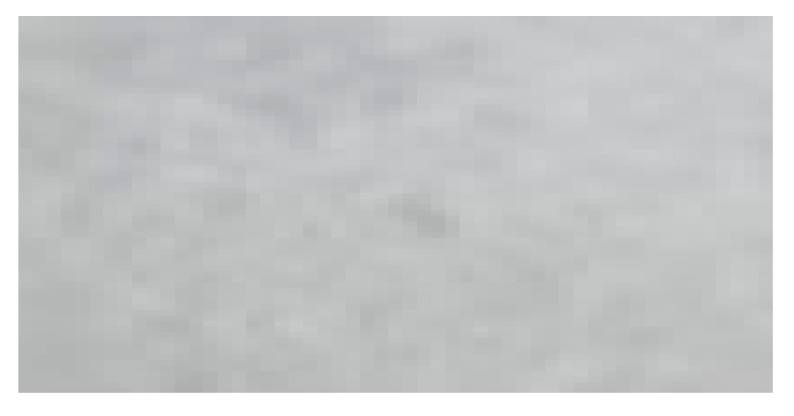	4.69 ± 0.90 ^a^	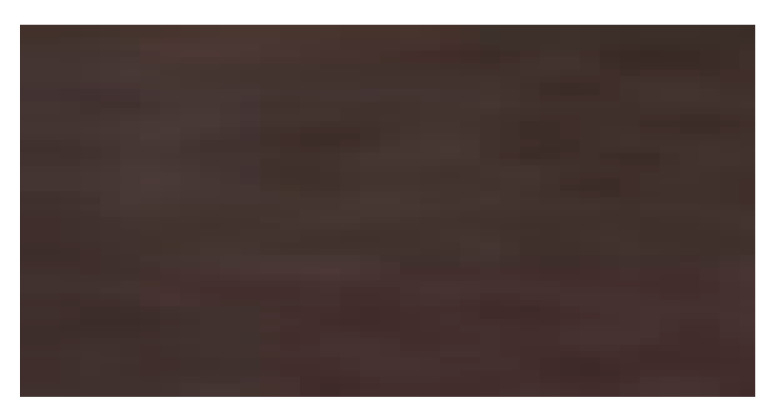	70.32 ± 0.99 ^a^	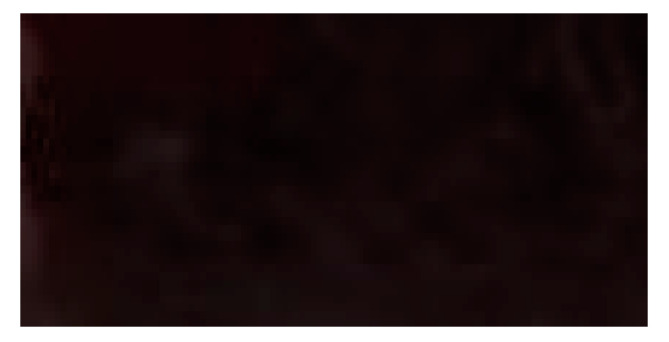	85.97 ± 2.03 ^a^	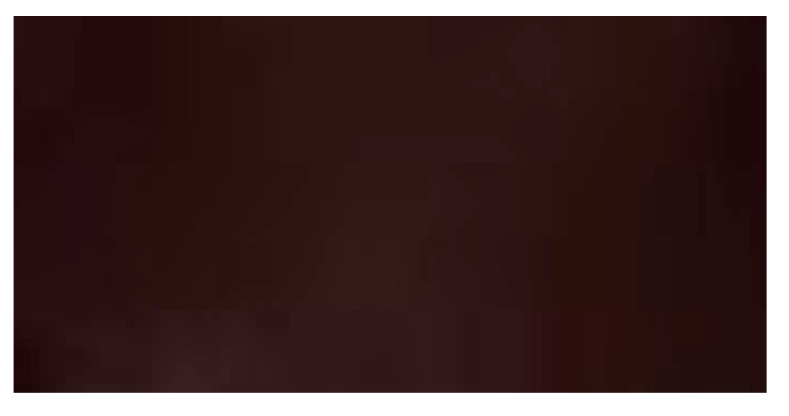	81.82 ± 0.30 ^a^

BAC, blueberry anthocyanin-derived cyanidin; G, gelatin; QC, quaternary chitosan. Values are given as mean ± SD (*n* = 3). * Different letters indicate significantly different values between the data (*p* < 0.05).

**Table 4 foods-13-02237-t004:** Changes in the acidity, pH value, and total bacterial count of milk and the color of QC-G-BAC films during the pasteurized milk storage.

Milk Exposure Time (h)	Milk Acidity (°T)	pH Value of Milk	Total Bacterial Count (Log_10_ CFU/mL) of Milk	QC-G-BAC0 Film		QC-G-BAC5 Film		QC-G-BAC10 Film		QC-G-BAC15 Film	
				Appearance	ΔE	Appearance	ΔE	Appearance	ΔE	Appearance	ΔE
0	15.5 ± 0.05 ^d^*	6.75 ± 0.04 ^a^	3.15 ± 0.14 ^d^	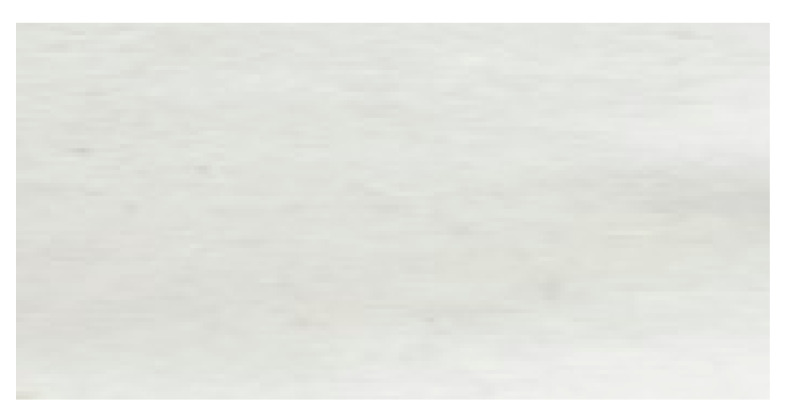	6.01 ± 0.14 ^a^	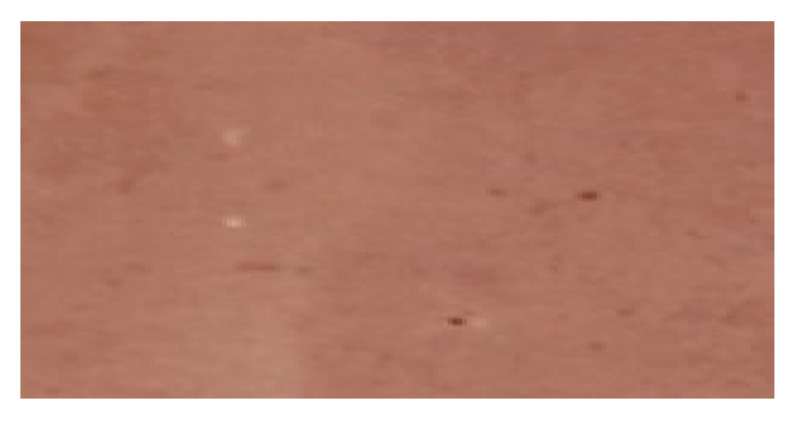	48.11 ± 0.84 ^a^	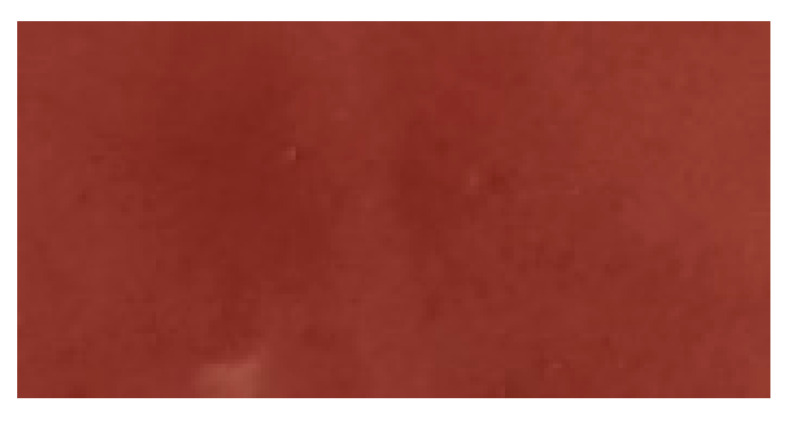	73.31 ± 0.61 ^a^	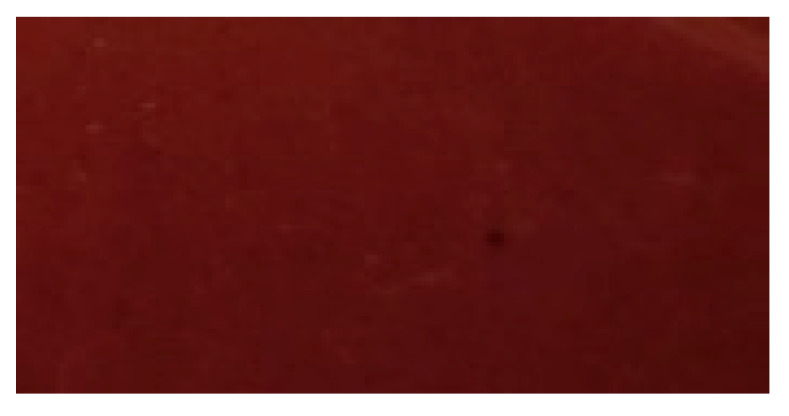	82.38 ± 0.43 ^a^
24	18.5 ± 0.95 ^c^	6.17 ± 0.37 ^b^	5.07 ± 0.38 ^c^	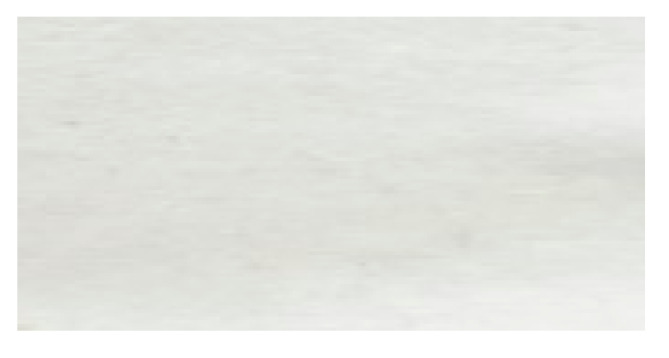	6.05 ± 0.34 ^a^	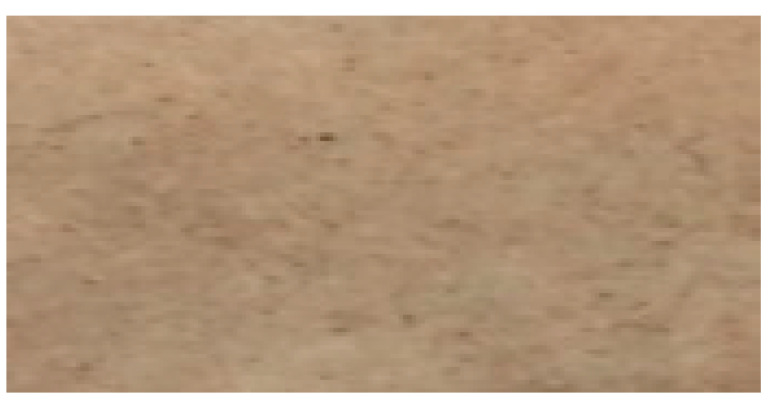	38.15 ± 0.94 ^b^	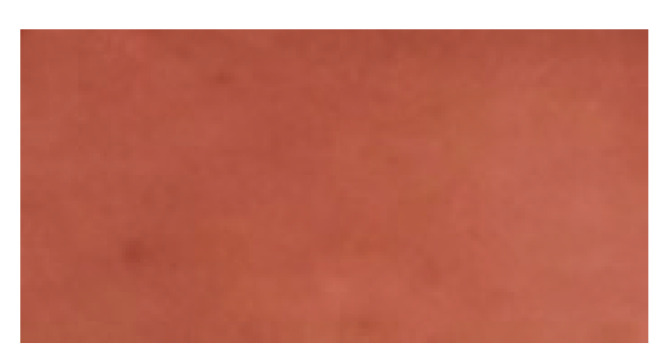	60.75 ± 0.94 ^b^	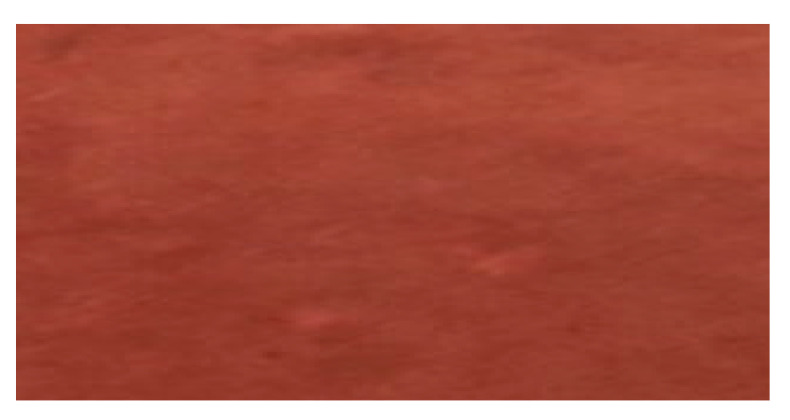	69.05 ± 0.69 ^b^
48	28.0 ± 1.10 ^b^	5.37 ± 0.12 ^c^	7.75 ± 0.28 ^b^	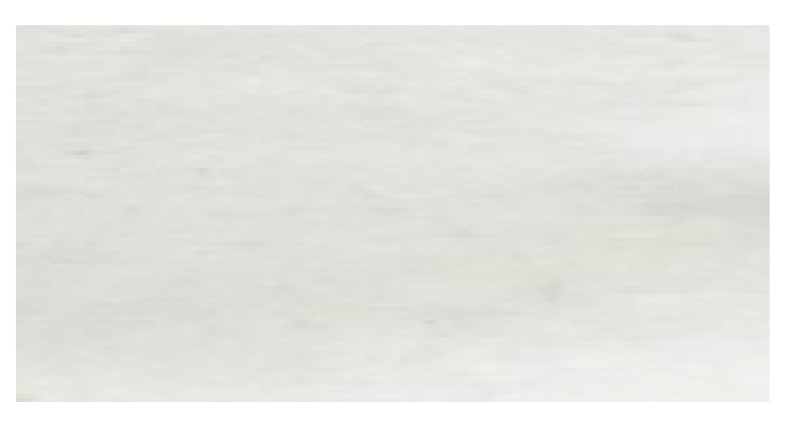	5.65 ± 0.30 ^a^	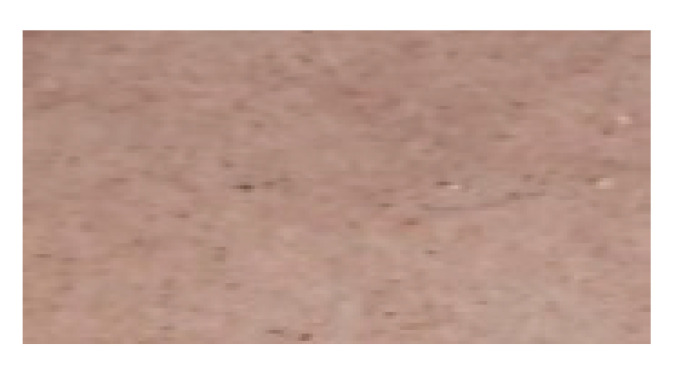	36.45 ± 0.64 ^c^	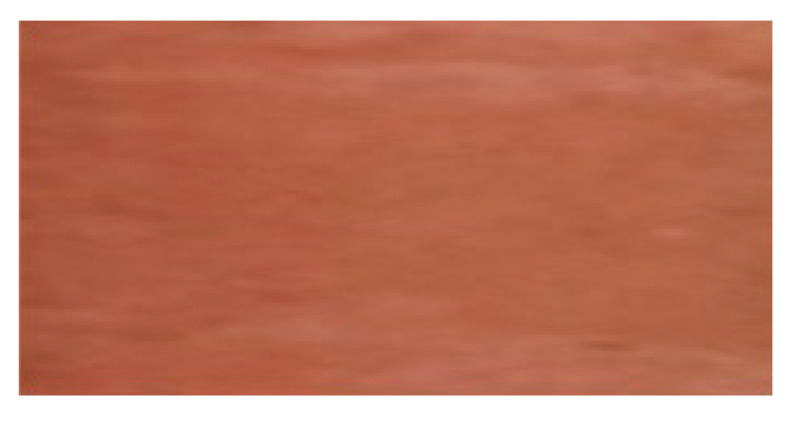	56.65 ± 0.95 ^c^	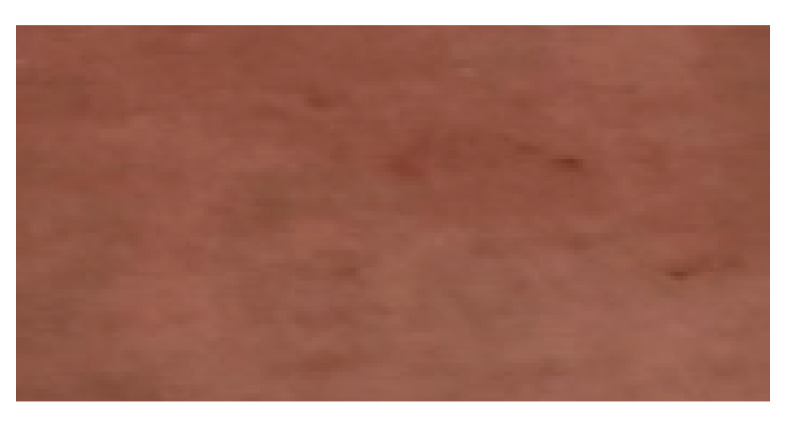	58.69 ± 0.55 ^c^
72	50.0 ± 1.40 ^a^	4.85 ± 0.08 ^d^	8.45 ± 0.34 ^a^	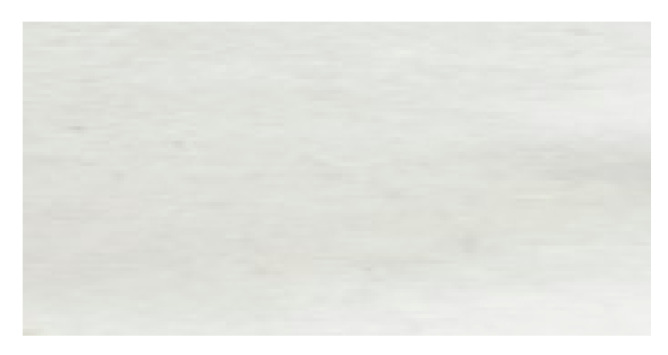	5.75 ± 0.27 ^a^	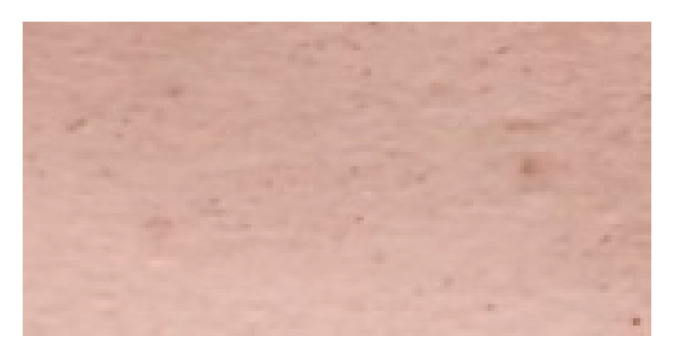	25.35 ± 0. 72 ^d^	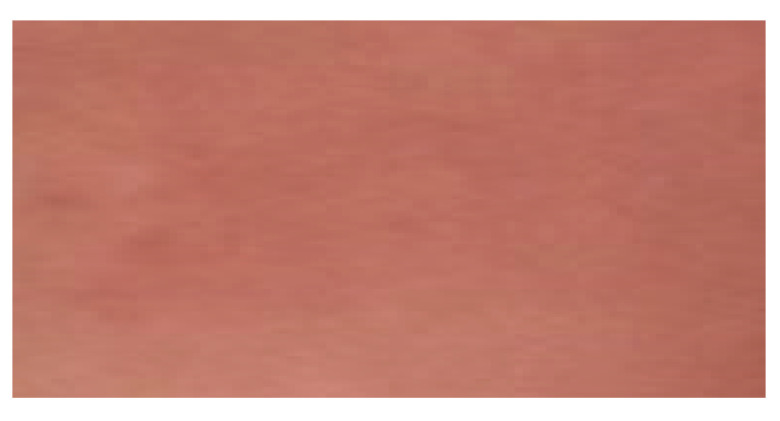	49.67 ± 0.22 ^d^	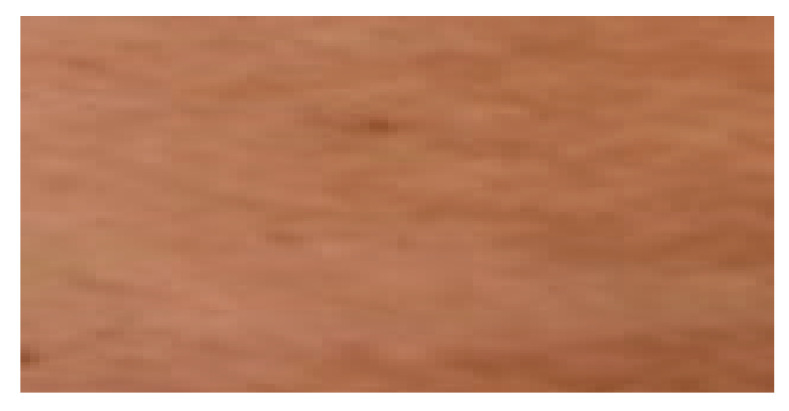	53.47 ± 0.47 ^d^

BAC, blueberry anthocyanin-derived cyanidin; G, gelatin; QC, quaternary chitosan. Values are given as mean ± SD (*n* = 3). * Different letters in the same column indicate significantly different values between the data (*p* < 0.05).

**Table 5 foods-13-02237-t005:** The Pearson correlation coefficient was calculated for the total color difference (ΔE) of the indicator films in relation to shrimp and milk quality parameters.

Food Items	Parameters	Pearson Correlation Coefficient			
		ΔE _QC-G-BAC0_	ΔE _QC-G-BAC5_	ΔE _QC-G-BAC10_	ΔE _QC-G-BAC15_
Shrimp	TVB-N level	−0.381	0.982 **	0.991 **	0.977 **
Milk	Acidity	−0.666	−0.930 *	−0.862	−0.853
	pH value	0.826	0.946 *	0.965 *	0.985 *
	Total bacterial count	−0.865	−0.908 *	−0.957 *	−0.992 **

* and ** indicate that the values are statistically significant at *p* < 0.05 and *p* < 0.01, respectively.

## Data Availability

The original contributions presented in the study are included in the article, further inquiries can be directed to the corresponding author.

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
