# Peer review of "Intelligent Food Packaging: Quaternary Ammonium Chitosan/Gelatin Blended Films Enriched with Blueberry Anthocyanin-Derived Cyanidin for Shrimp and Milk Freshness Monitoring"

_foods, 2024, doi:10.3390/foods13142237_

Round 1

Reviewer 1 Report

Comments and Suggestions for Authors

The manuscript foods-3094994 is well conceived: characterization of quaternary ammonium chitosan-gelatin films with the addition of cyanidin and their specific application to selected food products. However, I would point out several omissions/technical errors, the correction of which would contribute to the quality of the manuscript:

1. The title is too long. I suggest a simpler version:

Intelligent Food Packaging: Chitosan/Gelatin Films enriched with Blueberry Anthocyanin-Derived Cyanidin for Shrimp and Pasteurized Milk Freshness Monitoring

2. lines 14-15: check the units for tensile strength, it should be MPa, not %. Correlate to the equation you provided (line 136)

3. line 70: why did you choose exactly these two products? What was the selection parameter?

4. line 104: equilibrium of what? Please add.

5. lines 121, 136, 137, 146: Mark the equations with numbers in parentheses according to the order of appearance in the text. Check the author's guide.

6. lines 159-160: „...containing 15ml of 98% acetic acid solution and 1 mol/l ammonia water at 20oC…” Written like this, it turns out that the solutions were mixed? Or you had a container with acetic acid and another one with ammonia water?

7. lines 240-242: As shown in Figure 3, QC-G-BAC0 film showed 3279 cm-1 (N-H and O-H stretching), 2928 cm-1 (C-H stretching), 1638 cm-1 (C=O stretching) and 1542 cm-1 (N-H and C-N stretching) [16,23].” should be “As shown in Figure 3, QC-G-BAC0 film showed stretching at 3279 cm-1 (N-H and O-H), 2928 cm-1 (C-H), 1638 cm-1 (C=O) and 1542 cm-1 (N-H and C-N) [16,23].”

8. lines 246-249: Is this band near 3284 cm-1 the only example?

9. line 275: check if the statement “As shown in Figure 4B, the QC-G-BAC0 film has a high light transmittance, similar to that of chitosan-G film” really refers to reference 23 (She et al., Statins aggravate insulin resistance through reduced blood glucagon-like peptide-1 levels in a microbiota-dependent manner)

10. lines 304-308: provide more similar literature findings

11. lines 315-318: these 2 sentences seem to be in conflict. Please, rephrase this part to be clearer.

12. line 325: change to … ammonia gasses, and results were presented in Figure 6.”

13. figure 6: why are the intervals different (80 and 120 min)?

14. Section 3.5.: table 4 displays the results related to milk monitoring. The table is excellent, self-explanatory containing all the necessary information. Please present the shrimp monitoring results in the same way. It is also suggested to put the table in “landscape” orientation so that it would be easier to view/read it.

Author Response

Author's Reply to the Review Report

The manuscript foods-3094994 is well conceived: characterization of quaternary ammonium chitosan-gelatin films with the addition of cyanidin and their specific application to selected food products. However, I would point out several omissions/technical errors, the correction of which would contribute to the quality of the manuscript:

Comment 1: The title is too long. I suggest a simpler version: Intelligent Food Packaging: Chitosan/Gelatin Films enriched with Blueberry Anthocyanin-Derived Cyanidin for Shrimp and Pasteurized Milk Freshness Monitoring

Response 1: Thank you for pointing this out. We agree with this comment.

Comment 2. lines 14-15: check the units for tensile strength, it should be MPa, not %. Correlate to the equation you provided (line 136)

Response 2: Sorry for our negligence. The unit for tensile strength has now been corrected as MPa.

Comment 3. line 70: why did you choose exactly these two products? What was the selection parameter?

Response 3: Because milk and pork are the main foods that Chinese consumers currently use for protein supplementation, it is an industry with huge application demand.

Comment 4. line 104: equilibrium of what? Please add.

Response 4: The point being made here is: These films were then stored in a desiccator at 20 °C under a relative humidity of 50%. Misunderstood expression has been corrected.

Comment 5. lines 121, 136, 137, 146: Mark the equations with numbers in parentheses according to the order of appearance in the text. Check the author's guide.

Response 5: Thank you for pointing this out. Thanks for your critical suggestions and the manuscript has been modified as requested.

Comment 6. lines 159-160: „...containing 15ml of 98% acetic acid solution and 1 mol/l ammonia water at 20oC…” Written like this, it turns out that the solutions were mixed? Or you had a container with acetic acid and another one with ammonia water?

Response 6: The 2.5 cm × 2.5 cm film samples were separately attached to the headspace of a sealed container containing 15 mL of 98% acetic acid solution or 1 mol/L ammonia water at a temperature of 20 °C. Misunderstood expression has been corrected.

Comment 7. lines 240-242: „As shown in Figure 3, QC-G-BAC0 film showed 3279 cm-1 (N-H and O-H stretching), 2928 cm-1 (C-H stretching), 1638 cm-1 (C=O stretching) and 1542 cm-1 (N-H and C-N stretching) [16,23].” should be “As shown in Figure 3, QC-G-BAC0 film showed stretching at 3279 cm-1 (N-H and O-H), 2928 cm-1 (C-H), 1638 cm-1 (C=O) and 1542 cm-1 (N-H and C-N) [16,23].”

Response 7: Thank you for pointing this out. We agree with this comment.

Comment 8. lines 246-249: Is this band near 3284 cm-1 the only example?

Response 8: From the FT-IR spectrum, the most noticeable signal peak changes are at 1028 cm-1 (C-O-C), 3284 cm-1 and 1544 cm-1.

Comment 9. line 275: check if the statement “As shown in Figure 4B, the QC-G-BAC0 film has a high light transmittance, similar to that of chitosan-G film” really refers to reference 23 (She et al., Statins aggravate insulin resistance through reduced blood glucagon-like peptide-1 levels in a microbiota-dependent manner)

Response 9: Sorry for our negligence. The misquote has been corrected (Enhanced antioxidant activity of fish gelatin–chitosan edible films incorporated with procyanidin).

Comment 10. lines 304-308: provide more similar literature findings

Response 10: More similar literature findings are discussed.

Comment 11. lines 315-318: these 2 sentences seem to be in conflict. Please, rephrase this part to be clearer.

Response 11: Thanks for your vital suggestions and the wrong writing in the manuscript has been careful checked and re-written.

Comment 12. line 325: change to „… ammonia gasses, and results were presented in Figure 6.”

Response 12: Thank you for pointing this out. We agree with this comment.

Comment 13. figure 6: why are the intervals different (80 and 120 min)?

Response 13: This was a deficiency in our experimental design, which did not allow for uniform intervals when monitoring membrane color change. From the point of view of rigor, we will seriously consider your comments and improve them in the subsequent experiments.

Comment 14. Section 3.5.: table 4 displays the results related to milk monitoring. The table is excellent, self-explanatory containing all the necessary information. Please present the shrimp monitoring results in the same way. It is also suggested to put the table in “landscape” orientation so that it would be easier to view/read it.

Response 14: Thanks for your critical suggestions and the manuscript has been changed as requested.

Reviewer 2 Report

Comments and Suggestions for Authors

Comments to the Author
The authors of the manuscript foods-3094994-peer-review-v1, presented an interesting topic about Intelligent Food Packaging. The authors demonstrated proficiency in the use of different characterization techniques and the manuscript is well written.  However, some points need to be revised before publication.

  •        SEM analysis
  •  Isn’t  the observed cracks in SEM images of QC-G-BAC5 films and/or higher percentage degrade the quality of the prepared packaging material. Please explain?
  •  
  •   FTIR analysis
  • Page 7, line 239….replace the word “determine” with “explore”
  • Page 7, line 247….remove the word “significantly” as change in band strength was not significant as observed from Fig. 3 
  •  
  • Color and Light Transmittance
  •   Why light transmittance behavior of the QC-G-BAC15 was differ than that of QC-G-BAC5 and QC-G-BAC10 samples.  The QC-G-BAC15 sample exhibits the highest transmittance in the UV and visible range compared with other samples. The discussion given the manuscript don’t applicable of the sample QC-G-BAC15. Please explain.
  •  
  • Tabel 4
  • Please revise the font size of the first fourth columns and give appropriate title for these columns.

Author Response

Author's Reply to the Review Report 

Comments 1: SEM analysis. Isn’t the observed cracks in SEM images of QC-G-BAC5 films and/or higher percentage degrade the quality of the prepared packaging material. Please explain?

Response 1: We thank the reviewers for their comments. After reviewing the literature and repeating the experiments, it was found that the reason for the laminar cracks in the previously collected SEM patterns was that the samples waited too long in a high temperature environment before being examined, so the membranes dried out and cracked. The SEM patterns have been re-determined and re-analyzed.

Comments 2: FT-IR analysis. Page 7, line 239, replace the word “determine” with “explore”.

Page 7, line 247, remove the word “significantly” as change in band strength was not significant as observed from Fig. 3.

Response 2: Thanks for your critical suggestions and the manuscript has been changed as requested.

Comments 3: Color and Light Transmittance. Why light transmittance behavior of the QC-G-BAC15 was differ than that of QC-G-BAC5 and QC-G-BAC10 samples. The QC-G-BAC15 sample exhibits the highest transmittance in the UV and visible range compared with other samples. The discussion given the manuscript don’t applicable of the sample QC-G-BAC15. Please explain.

Response 3: Sorry for our negligence. We re-determined the transmittance of the films and found that the legend was mislabeled, which has now been corrected.

Comments 4: Tabel 4. Please revise the font size of the first fourth columns and give appropriate title for these columns.

Response 4: The font size of the first fourth columns in Tabel 4 has been adjusted as required and appropriate title for these columns has been given.

Reviewer 3 Report

Comments and Suggestions for Authors

This paper titled "Intelligent Food Packaging with Blueberry

Anthocyanin-Derived Cyanidin/Quaternary Ammonium

Chitosan/Gelatin Blended Film Prepared for Shrimp and

Pasteurized Milk Freshness Monitoring" is a very interesting paper and suitable for this journal. This paper has some shortcomings that must be taken into account by the authors, to make it more interesting. Therefore, I suggest the authors determine d-spacing by DRX, the Figure 4A does not seem to be real, there is no thermal analysis (TGA-DTGA and DSC), there is no biodegradability analysis of the films, there is no water solubility analysis of the films. Therefore, the authors are suggested to make these determinations. and suggestions to make the paper more attractive.

Author Response

Author's Reply to the Review Report 

Comments 1: This paper titled "Intelligent Food Packaging with Blueberry Anthocyanin-Derived Cyanidin/Quaternary Ammonium Chitosan/Gelatin Blended Film Prepared for Shrimp and Pasteurized Milk Freshness Monitoring" is a very interesting paper and suitable for this journal. This paper has some shortcomings that must be taken into account by the authors, to make it more interesting. Therefore, I suggest the authors determine d-spacing by DRX, the Figure 4A does not seem to be real, there is no thermal analysis (TGA-DTGA and DSC), there is no biodegradability analysis of the films, there is no water solubility analysis of the films. Therefore, the authors are suggested to make these determinations. and suggestions to make the paper more attractive.

Response 1: Thanks for your critical suggestions and the manuscript has been changed as requested. Following the reviewers' suggestions, the authors added water solubility analysis, DRX determination of d-spacing, thermal analysis (TGA-DTG) and biodegradability analysis of the films, as well as improving the quality of Figure 4A. Details are given in the revised manuscript.